# Intracellular sphingosine releases calcium from lysosomes

**Doris Höglinger[1], Per Haberkant[1], Auxiliadora Aguilera-Romero[2], Howard Riezman[2], Forbes D Porter[3], Frances M Platt[4], Antony Galione[4], Carsten Schultz[1]\***

[1]Cell Biology and Biophysics Unit, European Molecular Biology Laboratory, Heidelberg, Germany; [2]Department of Biochemistry, University of Geneva, Geneva, Switzerland; [3]Eunice Kennedy Shriver National Institute of Child Health and Human Development, National Institutes of Health, Bethesda, United States; [4]Department of Pharmacology, University of Oxford, Oxford, United Kingdom

**Abstract** To elucidate new functions of sphingosine (Sph), we demonstrate that the spontaneous elevation of intracellular Sph levels via caged Sph leads to a significant and transient calcium release from acidic stores that is independent of sphingosine 1-phosphate, extracellular and ER calcium levels. This photo-induced Sph-driven calcium release requires the two-pore channel 1 (TPC1) residing on endosomes and lysosomes. Further, uncaging of Sph leads to the translocation of the autophagy-relevant transcription factor EB (TFEB) to the nucleus specifically after lysosomal calcium release. We confirm that Sph accumulates in late endosomes and lysosomes of cells derived from Niemann-Pick disease type C (NPC) patients and demonstrate a greatly reduced calcium release upon Sph uncaging. We conclude that sphingosine is a positive regulator of calcium release from acidic stores and that understanding the interplay between Sph homeostasis, calcium signaling and autophagy will be crucial in developing new therapies for lipid storage disorders such as NPC.

*For correspondence: schultz@embl.de

**Competing interests:** The authors declare that no competing interests exist.

## Introduction

Sphingosine (Sph) and other small sphingolipids such as ceramide (Cer) and sphingosine 1-phosphate (S1P) are bioactive lipids, which play crucial roles in central cellular processes such as apoptosis, cell growth, differentiation and senescence (*Zheng et al., 2006*; *Bartke and Hannun, 2009*; *Hannun and Obeid, 2008*). While Cer and S1P have been extensively studied and reviewed (*Futerman and Hannun, 2004*; *Spiegel and Milstien, 2003*; *Alvarez et al., 2007*; *Pyne and Pyne, 2010*; *Arana et al., 2010*), only limited information on the signaling mechanisms or cellular targets of Sph is available (*Jefferson and Schulman, 1988*; *Smith et al., 1997*; *Jarvis et al., 1997*; *Chang et al., 2001*). In cells, Sph can be generated in lysosomes, the plasma membrane, the endoplasmic reticulum (ER) as well as the Golgi complex by hydrolysis of Cer through the action of ceramidases (*Bernardo et al., 1995*; *Hwang et al., 2005*; *Sun et al., 2008*; *Xu, 2006*; *Mao et al., 2001*). In a physiological context, Sph is reported to mediate cell growth arrest and apoptosis (*Cuvillier, 2002*; *Suzuki et al., 2004*) and is involved in the pathophysiology of the lysosomal storage disorder Niemann-Pick disease type C (NPC). In this progressive neurodegenerative disease, which is caused by mutations in the lysosomal proteins NPC1 or NPC2, Sph accumulates alongside other lipids such as cholesterol, sphingomyelin, and glycosphingolipids (*te Vruchte et al., 2004*). Specifically, Sph storage is the first detectable biochemical change following inactivation of NPC1 leading to chronically lowered calcium concentration in the acidic compartment, as well as subsequent

**eLife digest** Sphingosine is a small fat molecule that has been suggested to act as a signal inside cells. Individuals with a rare neurodegenerative disease called Niemann-Pick disease type C accumulate sphingosine and other fat molecules in cell compartments called lysosomes. Intriguingly, this fat accumulation is accompanied by an altered movement of calcium ions in and out of lysosomes.

In healthy cells, an increase in calcium ion levels can trigger a process called autophagy, in which proteins and other cell components are destroyed in a controlled manner. This is thought to be caused by the release of calcium ions from lysosomes, which stimulates a protein called TFEB to move into the nucleus of the cell to activate genes involved in autophagy. Two proteins on the surface of lysosomes called TPC1 and TPC2 are believed to act as channels that can release calcium ions from lysosomes. However, it was not clear how sphingosine could disrupt calcium ion movements in patients with Niemann-Pick disease type C.

Here, Hoeglinger et al. have used a new approach to understand how calcium ions and sphingosine are linked in both healthy and diseased cells. The experiments use a form of sphingosine called "caged sphingosine" that is only activated when it is exposed to a flash of light, which makes it possible to increase the levels of this molecule in cells in a precise way. Hoeglinger et al. found that sphingosine triggered the release of calcium ions from lysosomes. This release required the TPC1 protein and resulted in TFEB moving into the cell nucleus.

Further experiments confirm that sphingosine accumulates in the lysosomes of cells taken from patients with Niemann-Pick disease type C. In these cells, the activation of caged sphingosine resulted in a much smaller release of calcium ions from lysosomes than that observed in healthy cells. Together, Hoeglinger et al.'s findings show that sphingosine acts as a signal to trigger the release of calcium ions from lysosomes, which in turn promotes autophagy. The next challenge is to find out exactly how sphingosine opens the calcium ion channels.

secondary lipid storage (*Lloyd-Evans et al., 2008*). On the other hand, the calcium release from acidic stores was recently monitored by a genetically encoded calcium indicator directly fused to endolysosomes which showed reduced calcium release from NPC patient cells (*Shen et al., 2012*) in agreement with previous studies (*Lloyd-Evans et al., 2008*) but found that the calcium content in the lysosomes remains unchanged and that the calcium release is blocked by the accumulated lipids. This is in contrast to previous studies showing that the defect in calcium levels in NPC disease is likely due to a store filling defect and that intraluminal calcium levels were reduced using an intra-lysosomal calcium probe (*Lloyd-Evans et al., 2008*). Further research is necessary to fully elucidate the interplay between calcium uptake and release from the endolysosomes and how this mechanism is deregulated in the NPC disease.

Cells use calcium as a versatile tool for modulating intracellular signaling by increasing the intra-cellular free calcium concentration $[Ca^{2+}]_i$ either globally or locally. Second messengers such as *myo*-inositol 1,4,5-trisphosphate (IP$_3$) or cyclic ADP-ribose (cADPR) open well-characterized calcium channels in the endoplasmic reticulum (ER) or the muscle sarcomere, the main intracellular calcium store (*Berridge et al., 2003*; *Fliegert et al., 2007*). The intraluminal calcium concentrations in the ER varies from 100–800 µM (*Burdakov et al., 2005*). Since the endosomal/lysosomal system was identified as an important source of intracellular calcium (*Haller et al., 1996*), the luminal calcium concentrations of acidic vesicles was determined to range between 400–600 µM (*Christensen et al., 2002*). The mechanisms responsible for $Ca^{2+}$filling and release from these acidic stores are not yet fully understood. The main players involved in lysosomal calcium release and signaling are transient receptor potential (TRP) channels such as mucolipin 1 (TRPML1) (*Pryor et al., 2006*) and two two-pore channels (TPC1 and TPC2), which are thought to be $Ca^{2+}$ channels activated by nicotinic acid adenine dinucleotide phosphate (NAADP), $Mg^{2+}$or phosphatidylinositol 3,5-bisphosphate (PI(3,5)P$_2$) (*Brailoiu et al., 2009*; *Wang et al., 2012*; *Ruas et al., 2014*; *Brailoiu et al., 2010*; *Rybalchenko et al., 2012*; *Pitt et al., 2010*; *Pitt et al., 2014*; *Jha et al., 2014*; *Ruas et al., 2015*; *Jentsch et al., 2015*). A very recent study showed that lysosomal calcium signaling regulates autoph-agy through the actions of the phosphatase calcineurin and the transcription factor EB (TFEB)

(*Medina et al., 2015*) which places the lysosome at the center of this very important signaling hub and underlines the importance of understanding the regulation of lysosomal calcium signaling.

In this work, we investigate the role of Sph in intracellular calcium signaling by employing photo-activatable ("caged") Sph. Caged lipids are biologically inactive as long as they are covalently attached to certain photolabile groups ("cage groups"), which prevent recognition and metabolism inside cells. The active species is then released inside cells by short irradiation with a flash of light, which induces a cleavage reaction in the cage group (*Höglinger et al., 2014*). In this manner, it is possible to rapidly elevate the intracellular concentration of the lipid with precise spatiotemporal resolution. It is currently difficult to know the physiological and spatial distribution of sphingosine generated under physiological conditions, particularly inside the lysosome where the majority of sphingosine is generated via the action of acidic ceramidases. This tool therefore provides a very useful system with which to probe sphingosine cell biology in greater detail.

We discovered that uncaging Sph in a variety of cell types leads to an immediate and transient increase in cytosolic calcium, which is not released from the ER but from acidic stores. We further pinpoint the endosomal/lysosomal channel TPC1 to be a main contributor to this response and demonstrate that Sph-induced calcium release leads to the nuclear translocation of TFEB. We also show that the calcium release in NPC patient cells is reduced while the Sph levels in the lysosomal compartments are increased. These findings will be helpful in elucidating the pathogenic cascade of NPC and will generally further our understanding of the function of the acidic compartments in calcium homeostasis.

## Results and discussion

### Design and synthesis of caged Sph and caged dhSph

We synthesized caged variants of Sph by chemically attaching two different cage groups to the amino functionality of Sph (*Figure 1*): $R_1$ is a diethylamino-coumarin group, which allows for very fast, sub-second uncaging kinetics (*Schade et al., 1999*; *Hagen et al., 2003*) as well as visualization of cages lipids via its fluorescent properties. $R_2$ is a well-established, but non-fluorescent 4,5-dimethoxy-2-nitrobenzyl (NB) group (*Il'ichev et al., 2004*). In order to exclude potential artifacts caused by irradiation or cleavage of the cage groups, we also synthesized a negative control compound, coumarin-caged dihydrosphingosine (dhSph), a naturally occurring lipid that is structurally very closely related to Sph.

### Stability of Cou-caged Sph in cells

Next, we checked whether the caged Sph was taken up by cells and if it was stable in the cellular environment. To this end, HeLa cells were incubated with 2 µM Sph-Cou for different times and subjected to +/- UV uncaging. In +UV cases, the whole dish of labelled cells was exposed to UV light from a mercury arc source for a comparably long time of 2 min to ensure complete uncaging of all cells. Cellular lipids were then extracted and coumarin-containing lipids were visualized by thin layer chromatographic analysis (TLC, *Figure 1—figure supplement 1*). We showed that caged Sph was taken up by cells and that its photo-induced cleavage was successful inside the cells. This experiment also demonstrated that caged Sph did not participate in lipid metabolism during up to 60 min of incubation, as no additional fluorescent bands could be detected on the TLC plate. The drawback of this analysis was that only lipids equipped with a coumarin cage could be visualized via TLC. In order to gain a more accurate estimate of the uptake, stability, and successful uncaging of Sph-Cou as well as the negative control compound dhSph, we investigated control HeLa cells, Sph-Cou and dhSph-Cou treated HeLa cells by lipidomic analysis. *Figure 1—figure supplement 2* shows the relative amounts of Sph, S1P, dhSph and dhS1P detected in control, non-UV treated (caged) and UV-treated (uncaged) conditions for uncaging Sph-Cou as well as dhSph-Cou. The stability of the caged Sph (and caged dhSph) in the cells was confirmed because comparable levels of sphingoid bases were measured in control and caged conditions, indicating that no cleavage of the carbamate-linked cage group took place. Additionally, incubation with Sph-Cou or dhSph-Cou did not appear to perturb the long chain base homeostasis in the cells as all endogenous sphingoid bases showed similar concentrations compared to control conditions. Under the exaggerated illumination conditions used in the TLC and lipidomics experiments, Sph levels were found to increase 3.4 fold relative to control

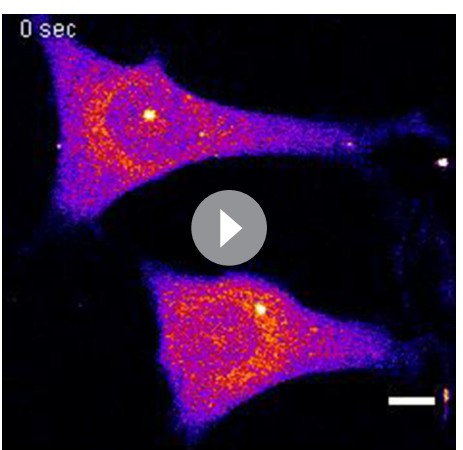

**Figure 1.** Structures of coumarin-caged sphingosine (Sph-Cou), nitrobenzyl-caged sphingosine (Sph-NB) and the negative control, coumarin-caged dihydrosphingosine (dhSph-Cou), respectively.
The following figure supplements are available for figure 1:

**Figure supplement 1.** Stability of caged Sph in cells.
**Figure supplement 2.** Comparative lipid analysis by mass spectrometry shows a successful uncaging reaction.

conditions in cells incubated with Sph-Cou, indicative of successful uncaging. This increase was specific for Sph and was not observed for dhSph (*Figure 1—figure supplement 2a*). The levels of S1P also increased in UV-treated conditions, albeit only at a minute level (0.2% of Sph). We attributed this increase to ongoing metabolism during and after uncaging reaction. It seems that the long time of illumination as well as time for the following steps (collection of the cells and extraction of the lipids) was long enough for phosphorylation to occur on a very small fraction of Sph (*Figure 1—figure supplement 2*). The negative control dhSph was also uncaged successfully, as indicated by a 13.9 fold increase of dhSph in uncaged cells compared to control (*Figure 1—figure supplement 2b*). This greater increase is explained by the fact that the caged lipids were added at the same concentrations, even though the endogenous levels of dhSph are 5 to 8 fold lower compared to Sph (according to our mass spectrometric data). Finally, it is important to consider that the TLC and mass spectrometric experiments were performed to show that the uncaging reaction is successful in a cellular environment. They should not be interpreted as quantitative measures of the amount of released Sph. Very long illumination conditions were necessary to uncage enough cells so that these analyses were made possible. All following live-cell experiments were performed on a dual scanner confocal microscope with local uncaging of a subcellular area in the cell with simultaneous fluorescence recording. The uncaging area and uncaging times could therefore be kept to a minimum. In this manner, a greatly reduced

**Video 1.** Local uncaging of Sph leads to calcium transients. Time-lapse movie of HeLa cells loaded with the calcium indicator Fluo-4 and treated with Sph-Cou. Cells were uncaged a t = 7 s for 3 s and the Fluo-4 fluorescence was recorded.

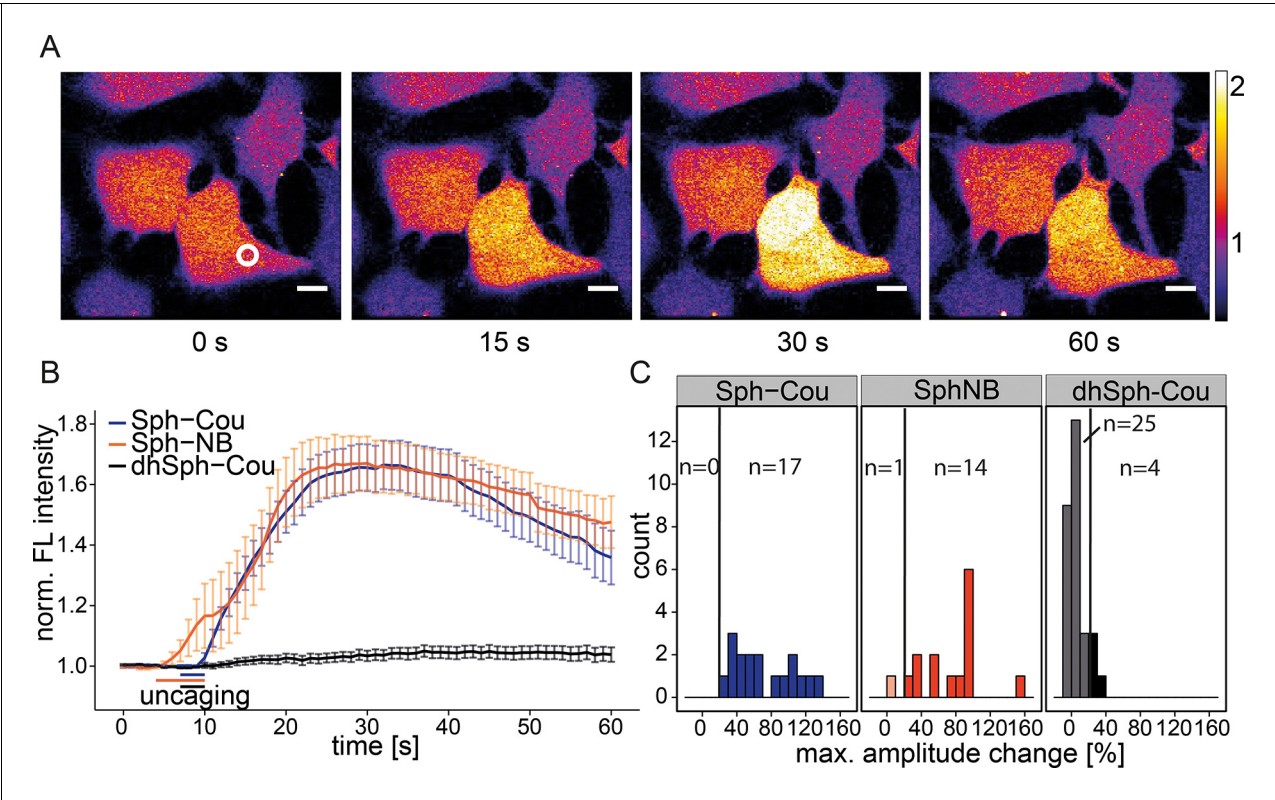

**Figure 2.** Local uncaging of Sph leads to calcium transients. (**A**) Time-lapse confocal microscopy images of HeLa cells loaded with the calcium indicator Fluo-4 and treated with Sph-Cou. Cells were irradiated within the white circle at t = 7 s for 3 s. (**B**) Mean Fluo-4 fluorescence traces of cells loaded with compounds Sph-Cou (17 cells), Sph-NB (15 cells), and dhSph-Cou (29 cells), respectively. Uncaging was carried out for 3 s for coumarin-caged compounds and for 6 s for nitrobenzyl-caged sphingosine. The duration of uncaging for each lipid is represented by the color-coded bars. Traces represent mean values with the standard error of the mean plotted as error bars. (**C**) Histogram showing the distribution of the maximum observed amplitude compared to baseline of each analyzed cell. The vertical line represents the threshold set at 20% amplitude increase for responding cells.

The following figure supplements are available for figure 2:

**Figure supplement 1.** Inhibition of Sph kinases.

**Figure supplement 2.** Free Sph does not induce calcium release.

release of Sph as well as minimal phosphorylation can be expected.

## Uncaging of Sph results in cytosolic calcium transients

In order to observe the immediate effects of photo-induced Sph release on intracellular calcium signaling in living cells, we loaded HeLa cells with the membrane-permeant calcium indicator Fluo-4/ AM (*Williams et al., 1999*), which reacts to increases in cytosolic calcium concentrations with corresponding increases in fluorescence intensity. We performed uncaging experiments using a confocal microscope by recording Fluo-4 fluorescence before and after brief irradiation (3 s–6 s) of a local circular area (~9 $\mu m^2$) within the cell. Cells reacted with an immediate (1–5 s), but transient increase in cytosolic calcium after uncaging either variant of caged Sph (Sph-Cou or Sph-NB, *Video 1* and *Figure 2a*). We calculated the mean calcium traces for each lipid and found a pronounced transient increase for both caged Sph variants, whereas the negative control compound dhSph-Cou gave no significant change in calcium levels under exactly the same delivery/uncaging conditions (*Figure 2b*), indicating that the signal was specifically due to sphingosine. To analyze single cell responses, we calculated the maximum amplitude of each calcium trace and visualized their distribution in a histogram (*Figure 2c*). We defined individual cells with more than 20% increase over the average baseline fluorescence as responding cells. While all cells responded when Sph-Cou/+UV treated, less than

15% (4 out of 29 cells) responded under control conditions (dhSph-Cou/+UV). Moreover, under control conditions, the few responding cells gave rise to markedly reduced amplitudes.

We then addressed the possibility whether S1P as a potential metabolite of the liberated Sph and known agonist of intracellular calcium release (*Meyer zu Heringdorf et al., 2003*), was essential for the calcium response. We performed uncaging experiments in the presence of two inhibitors of the sphingosine kinases: *N,N*-dimethylsphingosine (DMS) (*Edsall et al., 1998*) is a Sph analogue, which acts by competing for the binding of Sph. The second inhibitor, SKI-II acts as a mixed inhibitor of Sph and ATP binding (*Lim et al., 2012*). When performing uncaging experiments in presence of these inhibitors, we found only slight differences of the calcium response compared to basal conditions (*Figure 2—figure supplement 1*). These differences might be explained by cell-to-cell variability as well as a general perturbation of the cells when using these inhibitors. We conclude that the observed calcium signals were caused by the rapid, photoinduced Sph release rather than S1P formation. Interestingly, external addition of free Sph to the medium of the cells did not evoke changes in calcium levels (*Figure 2—figure supplement 2a*). The reasons for this are not fully resolved, but we speculate that Sph is more rapidly metabolized after entering cells than it is delivered. Sufficiently high doses of externally applied Sph (> 5 µM) are then cytotoxic. The rapid increase in Sph after uncaging overcomes this problem and hence leads to calcium signaling. We also added free Sph

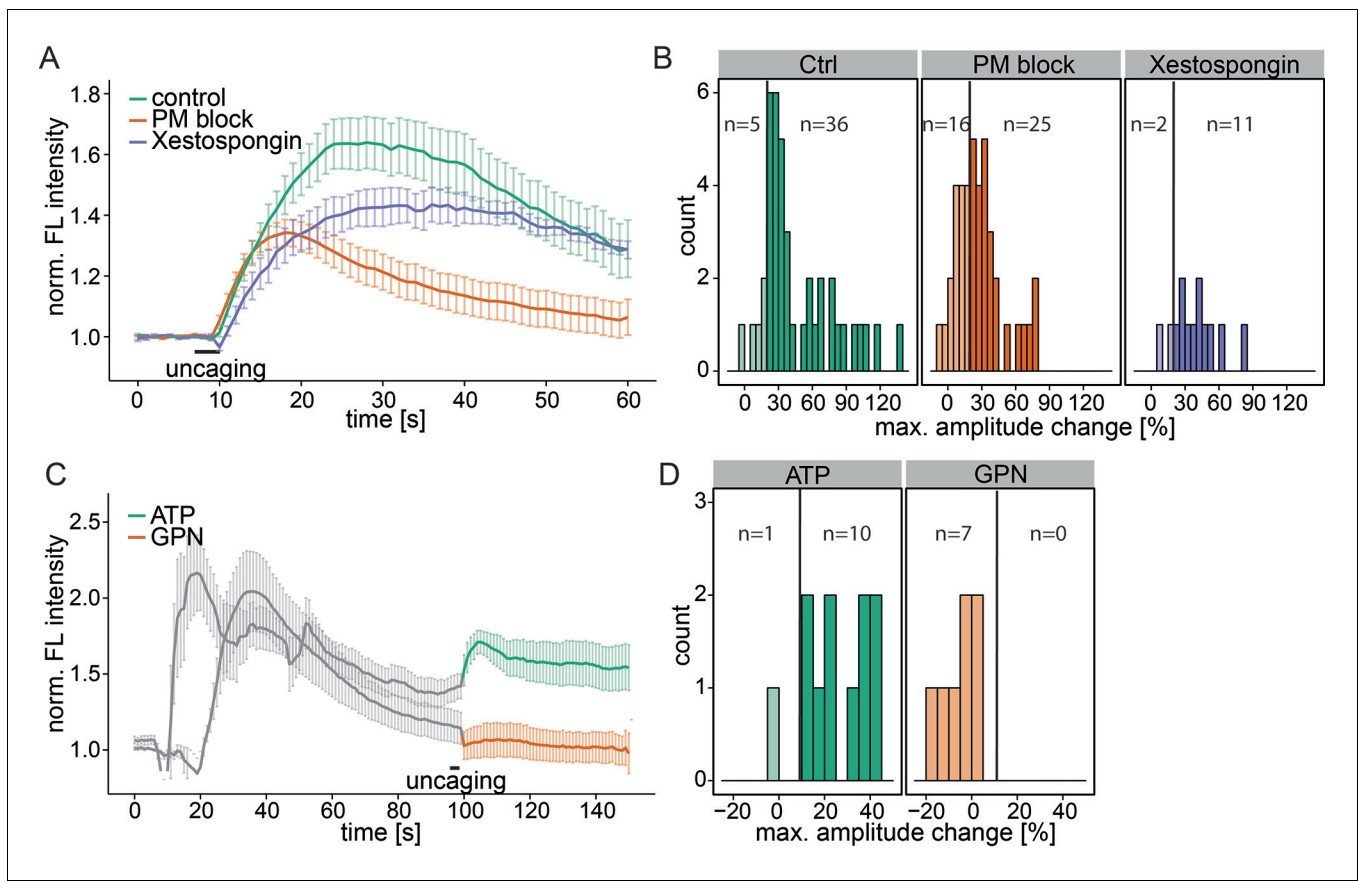

**Figure 3.** Investigating the source of Sph-induced calcium. (**A**) HeLa cells were loaded with 2 µM Sph-Cou in wild-type conditions (41 cells) or in conditions inhibiting plasma membrane channels by removing extracellular calcium with ethylene glycol tetraacetic acid (EGTA) and by inhibition with 5 mM $Ni^{2+}$ (PM block, 41 cells) or by blocking $IP_3$ receptors at the ER using Xestospongin C at 25 µM (13 cells). Traces represent the mean values with the standard error of the mean plotted as error bars. (**B**) Histograms showing the distribution of the maximum observed amplitude change compared to baseline of all cells, with a threshold at 20%. (**C**) HeLa cells were loaded with 2 µM Sph-Cou. After addition of either 10 µM ATP to stimulate release of ER $Ca^{2+}$ or addition of 200 µM glycyl-L-phenylalanine-beta-naphthylamide (GPN), which leads to release of $Ca^{2+}$ from the acidic stores through osmotic rupture, uncaging was performed at t = 100 s after the primary calcium transient had passed, as indicated by the bar. (**D**) Histograms showing the distribution of the maximum observed amplitude of the calcium increase after uncaging. Since these observed effects are second calcium transients, we lowered the threshold for responding cells to 10% amplitude change over baseline.

prior to the uncaging experiment and found again no difference in the calcium response. (*Figure 2—figure supplement 2b*).

## Investigating the origin of the calcium

To determine the source of the released calcium, we inhibited the plasma membrane calcium channels in an unspecific way by adding 5 mM $Ni^{2+}$ as well as ethylene glycol tetraacetic acid (EGTA) to the medium to complex all extracellular calcium (PM block). This inhibition did not block the Sph-induced calcium signal, although the mean amplitudes were reduced and 16 of 41 cells exhibited amplitudes below the 20% response threshold (*Figure 3a and b*). The initial (5 s) calcium increase was the same as for control conditions, demonstrating that plasma membrane calcium channels were not initially involved in this response. However, blocking PM channels leads to reduced ER calcium and thereby reduces calcium-induced calcium release from the ER. This likely accounts for the reduced amplitudes and response rates in PM block conditions.

To investigate the contribution of the ER as the major intracellular calcium store, we inhibited the inositol trisphosphate receptor with Xestospongin C (*Gafni et al., 1997*). The calcium signals still persisted with the same initial calcium increase and only slightly reduced amplitudes, suggesting that $IP_3$ channels were not directly involved (*Figure 3a and b*).

As a complementary approach, we stimulated the release of ER calcium by adding ATP and monitored the increase of cytosolic calcium via G-protein coupled receptor stimulation (*Figure 3c*). After the ATP-induced calcium transient had passed, we uncaged Sph and observed a second increase in calcium. Since the ER had been previously emptied, we reasoned that this new signal must originate from a different intracellular calcium store. Also, this result further strengthens our hypothesis that liberated Sph conversion to S1P, which is known to induce calcium release from the ER (*Mattie et al., 1994*), is not critical for the effect we observe with caged Sph.

To determine whether the acidic compartment calcium stores are involved, we used glycyl-L-phenylalanine-*beta*-naphthylamide (GPN), an agent which is hydrolyzed by cathepsin C in lysosomes and causes osmotic lysis of acidic vesicles, consequently releasing their content. When GPN was added to the cells, we observed an increase in cytosolic calcium, but uncaging Sph after this transient had passed failed to induce a second calcium release. Taken together, these results point towards the acidic compartment as the primary source of the Sph-induced calcium signal that we observed.

## Investigating the mechanism of acidic compartment calcium release by Sph

To further specify the machinery involved in this Sph-induced calcium efflux from the acidic compartments, we employed mouse embryonic fibroblasts (MEFs) derived from mice with a single knock out of either the endosomal/lysosomal calcium channels TPC1 or TPC2 as well as MEFs from a double knock out mouse (*Ruas et al., 2014*). In WT MEFs, uncaging of Sph led to an immediate cytosolic calcium increase in the same way as was observed in HeLa cells, indicating that this effect is not cell-type specific (*Figure 4a*). Due to the different morphology and size of the fibroblasts, the amplitudes of the calcium transients were less pronounced in MEFs than in HeLa cells, since the area of uncaging was kept identical. This prompted us to set the threshold for responders to 10% increase over baseline. TPC2 knockout MEFs also gave rise to calcium transients with comparable amplitudes. Employing TPC1 knockout fibroblasts, however, resulted in a large reduction of the mean calcium response. Under these conditions, only 6 of 25 cells exhibited amplitudes greater than the response threshold and even those were markedly decreased compared to WT conditions (*Figure 4a and b*). This striking effect shows the specificity of Sph-induced calcium release via the TPC1 channel. To confirm this, we also employed MEF from the TPC1/TPC2 double knockout mouse. Under these conditions, only 9 of 31 cells gave calcium signals with greater amplitudes than the threshold, much like in the TPC1-KO conditions. We conclude that there is a connection between an increase in cellular Sph and the release of calcium from the acidic compartments through the action of TPC1, which is known to be localized to early, recycling, and late endosomes as well as lysosomes (*Brailoiu et al., 2009*).

We also investigated another important acidic store calcium channel, TRPML1, with respect to Sph-induced calcium efflux. We used a fibroblast cell line derived from a patient suffering from mucolipidosis type IV (MLIV) (*Sun, 2000*). This disease is characterized by a loss-of-function mutation

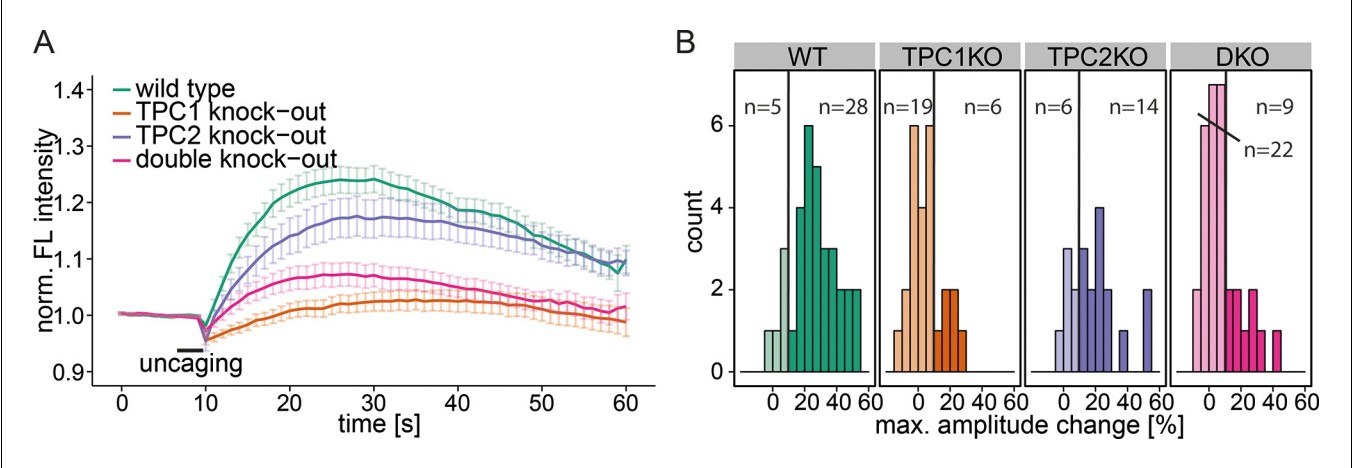

**Figure 4.** Knock-out studies of two-pore channels. (**A**) Primary mouse embryonic fibroblasts derived from a wild-type mouse (33 cells) or from two-pore channel 1 (TPC) knock-out mice (25 cells), two-pore channel 2 (TPC2, 20 cells) knock-out mice or from a double knock-out mouse (DKO, 31 cells) were loaded with 2 µM Sph-Cou and uncaged for 3 s as indicated. Traces represent the mean values with standard errors of the mean plotted as error bars. (**B**) Histograms showing the distribution of the maximum observed amplitude compared to baseline, with the threshold set to 10% increase over baseline.

The following figure supplements are available for figure 4:

**Figure supplement 1.** Contribution of the lysosomal calcium channel TRPML1.

**Figure supplement 2.** NAADP antagonist Ned-19.

in the acidic store calcium channel TRPML1 (*Bargal et al., 2000*). Again, control human fibroblasts gave an immediate calcium response upon release of Sph (*Figure 4—figure supplement 1*). MLIV patient fibroblasts also responded with an immediate and even higher calcium response suggesting that the observed effect is specific to TPC1 and that a loss of TRPML1 function does not interfere with Sph-induced calcium signaling. In fact, the loss of TRPML1 might even increase the importance of calcium efflux through TPCs as the only remaining way of maintaining acidic compartment calcium concentrations. Since two-pore channels are known to be critically involved in NAADP-dependent calcium signaling (*Brailoiu et al., 2009*; *Tugba Durlu-Kandilci et al., 2010*), we also tested the effect of a pharmacological antagonist of NAADP, Ned-19 (*Naylor et al., 2009*), on Sph-induced calcium release in HeLa cells (*Figure 4—figure supplement 2*). Ned-19 treated HeLa cells did not exhibit a markedly different calcium response after Sph uncaging, which is in line with previous observations that TPC1 is unresponsive to Ned-19 (*Pitt et al., 2014*). Several questions still remain on the interface between Sph and TPC1. It is unclear if there is direct interaction or if there are additional, yet unidentified mediators in this response. Also, the interaction with other TPC1 ligands such as NAADP needs to be investigated.

## Calcium signaling in Niemann-Pick disease type C

Niemann-Pick disease type C is marked by deregulated calcium homeostasis in the acidic compartments as well as an accumulation of Sph (*Lloyd-Evans et al., 2008*). We investigated Sph-dependent calcium signaling in this disease by performing uncaging experiments on human fibroblast cell lines derived from a healthy subject (control) and from a patient suffering from NPC1. Control cells reacted with an immediate increase in cytosolic calcium, showing again the broad applicability of these caged compounds across different cell types (*Figure 5a and b*). In these cells, we also reduced the response threshold to 10% due to the size of the fibroblasts compared to HeLa cells. NPC1 patient cells have been shown to have reduced calcium levels in the acidic stores due to a store filling defect (*Lloyd-Evans et al., 2008*). Using our setup, we could confirm that calcium transients in NPC1 cells after Sph uncaging exhibited significantly lower amplitudes (*Figure 5a and b*). This observation corroborated our finding that Sph releases calcium from lysosomes and is consistent

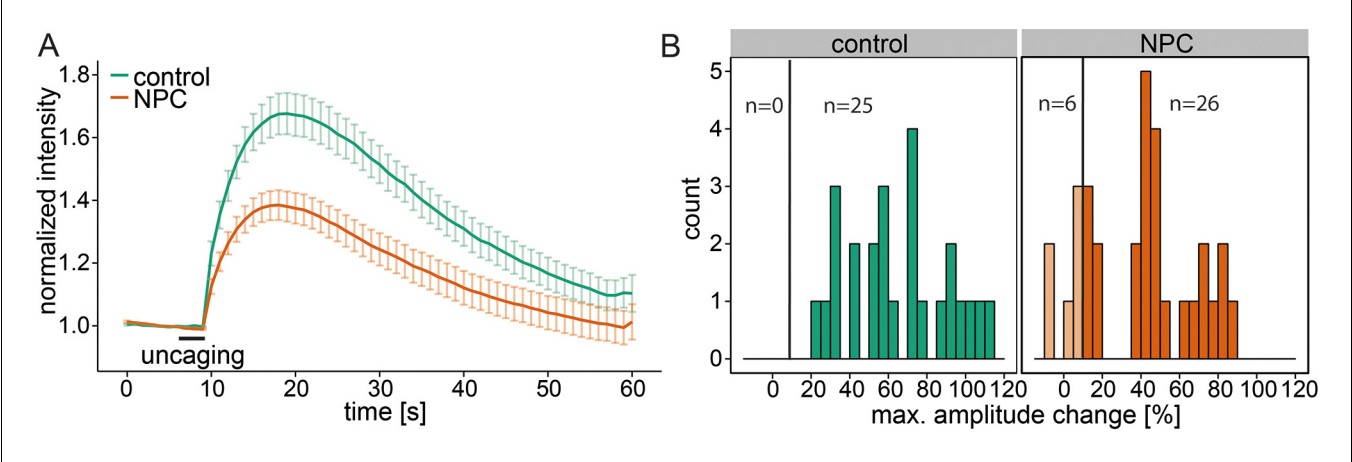

**Figure 5.** Calcium signaling in NPC disease. (**A**) Human fibroblasts derived from healthy subjects (control, 25 cells) or patients with Niemann Pick disease type C (NPC, 31 cells) were loaded with 5 µM Sph-Cou (1) and uncaged for 3s as indicated. Traces represent mean values with the standard error of the mean plotted as error bars. (**B**) Histograms showing the distribution of the maximum observed amplitude compared to baseline of all cells with the response threshold set to 10% increase over baseline.

with other studies (*Lloyd-Evans et al., 2008*) showing that in NPC disease these vesicles are filled with less calcium compared to control. It should be mentioned that this is in contrast to other findings in (*Shen et al., 2012*).

## Subcellular localization of Sph in NPC disease

Next, we investigated whether Sph is localized to the endosomal/lysosomal compartment in control as well as NPC1 fibroblasts by using a newly reported photoactivatable and clickable version of Sph (pacSph) (*Haberkant et al., 2015*). This Sph analogue is equipped with two modifications on the hydrophobic tail: a photo-crosslinkable diaziridine moiety and a functionality that allows for click reactions, used for staining Sph in cells with a fluorophore after fixation (*Figure 6a*). Using pacSph we could, for the first time, visualize the subcellular localization of Sph in individual control and NPC1 fibroblasts. After a short (10 min) pulse of 4 µM pacSph, it was localized in endosomal and lysosomal vesicles in both cell types (*Figure 6b*) as confirmed by co-localization studies with a LAMP1 antibody (*Figure 6—figure supplement 1*) and quantification by Pearson's correlation coefficient (*Figure 6c*). In NPC fibroblasts, these vesicles were bigger and much more numerous than under control conditions. They also exhibited much brighter overall fluorescence, indicative of higher cellular Sph concentrations. It has already been shown that NPC1 disease fibroblasts exhibit both lower acidic compartment calcium levels as well as an increased Sph storage (*Lloyd-Evans et al., 2008*).

Our findings strengthen the link between Sph and lysosomal calcium homeostasis. Additionally, we followed the metabolism of pacSph in both cell lines by performing pulse chase experiments, fixing and staining cells at different time points. In control fibroblasts, the endosomal/lysosomal staining almost completely disappeared after a chase of only 10 min, as represented by a drop in the Pearson's coefficient (*Figure 6c*). This corresponds to the rapid efflux and transport of Sph to other cellular compartments for further metabolism (*Hannun and Obeid, 2008*). In contrast, NPC fibroblasts still showed a marked endosomal/lysosomal staining with high fluorescence intensities after 10 min, likely due to failed or diminished efflux from these compartments in the disease. These localization studies further support our hypothesis that elevated endosomal/lysosomal Sph concentrations and reduced calcium levels are closely connected.

## Transcription factor EB (TFEB) translocation

Lysosomal calcium release was recently shown to lead to activation of calcineurin and subsequent nuclear translocation of transcription factor EB (TFEB), a master regulator of lysosomal biogenesis and autophagy (*Medina et al., 2015*). To test if Sph-induced calcium release also reproduces this

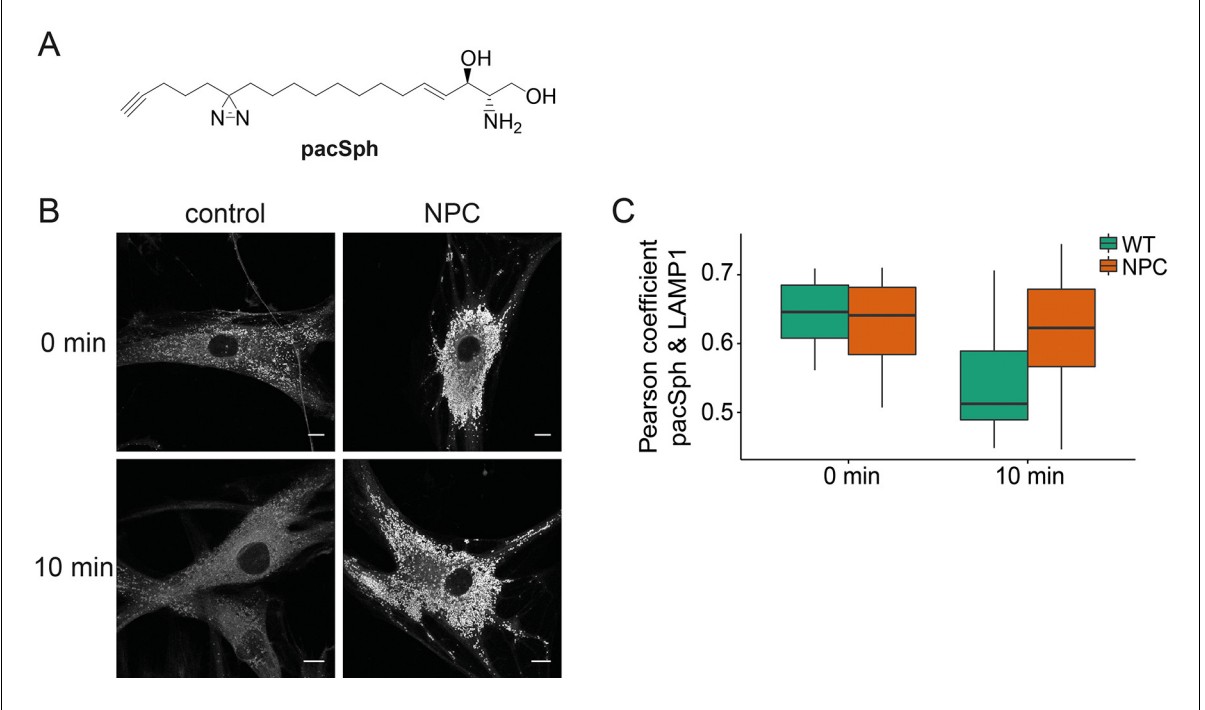

**Figure 6.** Subcellular localization of Sph. (**A**) Structure of pacSph. (**B**) Sph distribution in control and NPC human fibroblasts. Cells were incubated with 4 μM pacSph for 10 min, washed and either immediately photo-crosslinked and fixed (0 min) or incubated for further 10 min in buffer before crosslinking and fixation. Visualization was achieved by clicking Alexa488-azide to the terminal alkyne bond of pacSph. Confocal analysis shows a striking accumulation of Sph in the late endosomes/lysosomes of the NPC fibroblasts. For co-staining with the lysosomal marker LAMP1, see *Figure 6—figure supplement 1*. (**C**) Quantification of co-localization analysis by calculating Pearson's correlation coefficient for >6 cells in each condition.

The following figure supplement is available for figure 6:

**Figure supplement 1.** Co-localization with lysosomal markers.

effect, we performed uncaging experiments in HeLa cells overexpressing TFEB-GFP. Indeed, uncaging Sph and the subsequent calcium release (as monitored by the genetically encoded red calcium sensor, R-GECO (*Zhao et al., 2011*)) induced rapid translocation of TFEB-GFP to the nucleus (*Figure 7a*) in 14 out of 19 cells. This was not the case when calcium was released by directly uncaging caged calcium to a similar extent (*Figure 7b*). Only one in 11 cells showed a slight translocation of TFEB to the nucleus in the 15 min timeframe of the experiment. This further supports the findings that vital cellular signaling pathways are distinctly initiated on the lysosome surface. Medina et al. (*Medina et al., 2015*) found the mucolipin channel TRPML1 to be involved in activating calcineurin and translocating TFEB, while our findings suggest a contribution of TPC1. This could suggest a certain redundancy in lysosomal calcium release and requires further attention. It can be speculated that reduced calcium levels in NPC patients might fail to properly regulatetranscription through TFEB and that the subsequent differences in lysosomal maintenance and autophagy couldcontribute to the NPC phenotype.

## Conclusion and outlook

In summary, we have developed a method to increase the concentration of Sph in living cells using light in a spatially and temporally precise way. Thus, we interfered with sphingolipid metabolism minimally and were able to investigate resulting effects of elevated sphingosine levels. Dihydrosphingosine, although structurally very similar, served as a good negative control compound and underlined the high structural specificity of the interacting partners of Sph. We applied these tools to a variety of cell types, ranging from cultured HeLa cells to primary mouse embryonic fibroblasts and

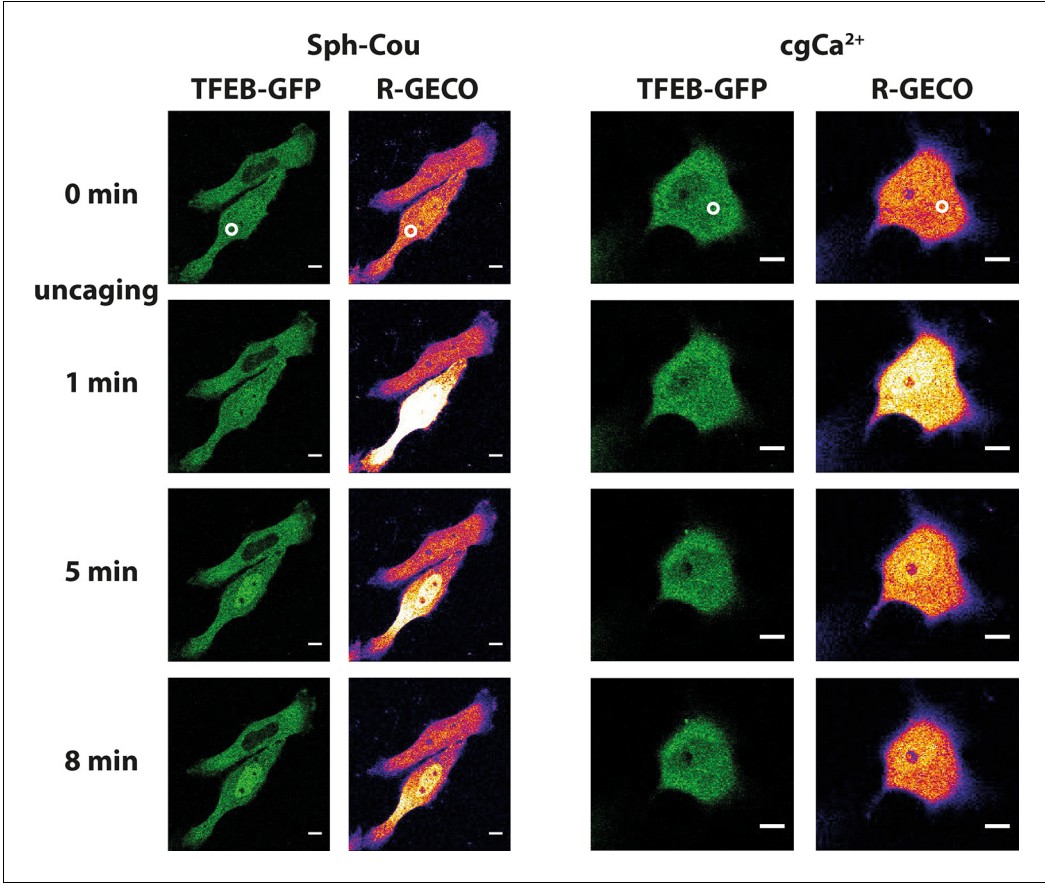

**Figure 7.** Sph uncaging leads to TFEB translocation to the nucleus. Time-lapse confocal microscopy images of HeLa cells transfected with TFEB-GFP and R-GECO (a genetically encoded calcium sensor) and loaded with either Sph-Cou or caged calcium (o-nitrophenyl EGTA/AM). Uncaging was performed in a small area within the cell for 3 s for Sph-Cou (lower cell) and 2 s for caged calcium.

human patient fibroblasts and showed that in all those cells, an acute increase in Sph was always followed by an immediate release of calcium from GPN-sensitive acidic stores. We also showed that plasma membrane and ER calcium channels were not initially involved in this calcium response. Furthermore, we could pinpoint TPC1, an endosomal/lysosomal channel of the two-pore channel family to be essential in this newly found signaling pathway. When employing embryonic TPC1 knockout fibroblasts, no calcium increase upon uncaging of Sph was observed. TPC2 knockout fibroblasts, however, reacted similarly to WT fibroblasts, confirming the specific requirement of the TPC1 isoform for Sph induced calcium signaling. The involvement of the other main acidic compartment calcium channel TRPML1 was also tested. Loss-of-function in TRPML1 led to a slight increase in Sph-induced calcium release, which might be explained by some redundancy between TRPML1 and TPCs in the regulation of lysosomal calcium concentrations.

Acidic compartment calcium signaling has recently emerged as an important part of intracellular events such as vesicle fusion and secretion (*Patel and Docampo, 2010*) as well as induction of autophagy and lysosomal biogenesis (*Medina et al., 2015*). Our knowledge of the machinery involved and its regulation, however, remains incomplete. The two-pore channels (TPCs) are the subject of current controversies regarding ion selectivity and dependencies on activating ligands. While many reports agree that TPC channels are NAADP-regulated $Ca^{2+}$ channels (*Brailoiu et al., 2009*; *Brailoiu et al., 2010*; *Pitt et al., 2010*; *Calcraft et al., 2009*), others indicate a broader ion specificity of TPCs (*Wang et al., 2012*; *Rybalchenko et al., 2012*; *Peiter et al., 2005*) or alternative ligands such as PI(3,5)P$_2$ (*Wang et al., 2012*; *Cang et al., 2013*). Another study hints towards a convergent regulation of TPC2 by $Mg^{2+}$, NAADP, PI(3,5)P$_2$ and two protein kinases (JNK and p38) (*Jha et al.,*

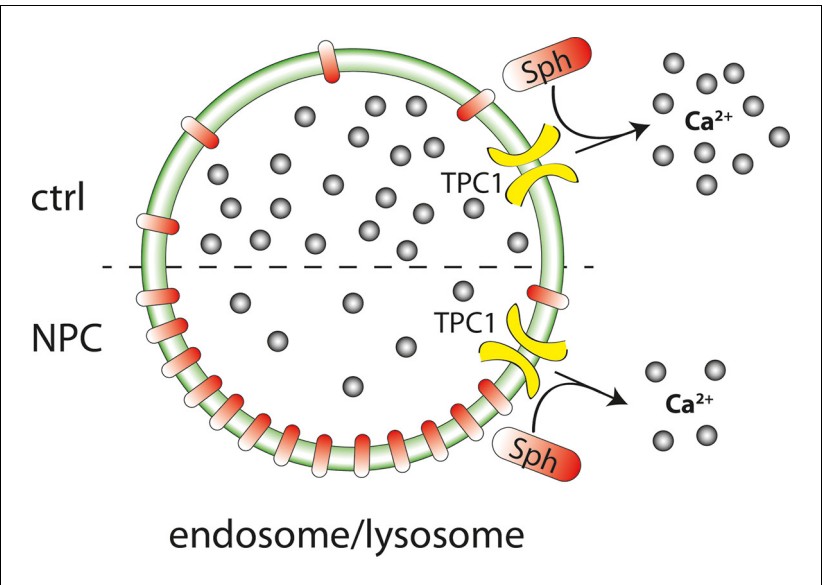

**Figure 8.** Schematic summary of findings. Sph causes calcium release from acidic stores through the action of TPC1. In Niemann-Pick disease type C, lower endolysosomal calcium levels occur together with increased Sph concentration. Stimulating lysosomal calcium release by uncaging Sph in NPC patient cells results in lower calcium amplitudes compared to cells from control subjects.

*2014*). A very recent study confirmed that TPCs conduct both $Ca^{2+}$ and $Na^+$ ions and that they are activated by NAADP as well as PI(3,5)P$_2$ (*Ruas et al., 2015*; *Jentsch et al., 2015*). The biological relevance of two-pore channels was recently highlighted by a report showing that TPCs are crucial to successful infection of host cells by the Ebola virus and that disruption of TPC function leads to halted virus trafficking (*Sakurai et al., 2015*). Interestingly, the progression of Ebola infection is also dependent on the NPC1 protein, which was shown to be necessary for viral escape from the lysosomal compartment *(Herbert, et al., 2015*, *Carette, et al., 2011)*. These recent findings will certainly encourage further efforts to understand the link between lysosomal calcium signaling and the defects in Niemann-Pick disease. In our study, we could confirm that NPC patient fibroblasts responded with lower amplitudes of the calcium transients after Sph uncaging, which is indicative of a reduced pool of calcium in the acidic compartments of these cells, in accordance with other studies (*Lloyd-Evans et al., 2008*; *Shen et al., 2012*).

*Figure 8* summarizes our results and illustrates that we pinpointed TPC1 as the mediator of a Sph induced rise in cytosolic calcium. We also correlated a reduced calcium release in fibroblasts from NPC patients with a significant storage of Sph in the late endosomes/lysosomes of these patient cells by using a novel bifunctional Sph probe. Short pulse-chase experiments also indicated lack of efflux from lysosomes in NPC fibroblasts compared to cells from healthy controls.

The question whether the NPC1 protein is directly or indirectly involved in the efflux of lipids, specifically Sph, from the acidic compartments has remained a controversial topic. NPC1 has been shown to be involved in the transport of amines (*Kaufmann and Krise, 2008*), but conclusive data on the transport of Sph and other potential substrates are absent as there is no direct functional assay for NPC1. A study using radiolabelled sphingolipids to follow metabolism and transport in NPC disease concluded that Sph efflux from endolysosomes is not restricted in NPC (*Blom et al., 2012*). However, these data were obtained using long, 1–4 h chase times during which the precursor lipid was likely metabolized to a high degree and incorporated into other lipids, leaving only a small fraction of Sph for quantitative analysis. This and other obstacles to the study of lipids affirm the need for new tools, which circumvent the lipid metabolic machinery and give a faithful representation of the subcellular lipid localization. With our tools and shorter chase times in the range of 10–20 min, we observed reduced acidic compartment calcium release as well as increased Sph storage and a failure in Sph efflux in NPC disease cells.

Disrupted autophagy is another hallmark of NPC shared by many other neurodegenerative disorders as neurons are particularly vulnerable to defects along the autophagic pathway (*Nixon, 2013*). TFEB was shown to up-regulate transcription of genes which support autophagosome formation and lysosomal biogenesis (*Settembre et al., 2011*) and could therefore be a considered a novel target to promote lysosomal clearance in lipid storage and neurodegenerative disorders. TFEB overexpression has already been shown to be beneficial in attenuating the storage phenotype in several storage disorders such as Batten disease (*Medina et al., 2011*) or Pompe disease (*Spampanato et al., 2013*). Similar overexpression strategies showed promising results even in mouse models of more common neurodegenerative disorders, such as Parkinson disease (*Decressac et al., 2013*), Alzheimer disease (*Polito et al., 2014*) or Huntington disease (*Tsunemi et al., 2012*). The regulatory functions of Sph and TFEB are therefore key targets for future investigations of neurodegenerative disorders.

Taken together, we describe intracellular Sph as a new small molecule effector of lysosomal calcium homeostasis, the lysosome as triggerable calcium store via the TPC1 channel, and specific transcription factor translocation as a consequence of lysosomal calcium release. Our results further highlight the importance of elucidating the interplay between lysosomal calcium signaling and Niemann-Pick type C disease. This interplay might also be important for the progression of diseases caused by Ebola or Marburg virus infection. Sph should be considered as a new and important player in this field and understanding the regulation of its signaling will likely further our understanding of these named diseases.

## Materials and methods

### Materials

Common laboratory chemicals were purchased from commercial sources (Acros, Fluka, Merck, Sigma-Aldrich or VWR) at highest available grade and were used without further purification. D-erythro sphingosine and D-erythro dihydrosphingosine, the sphingosine kinase inhibitor SKI-II, Xestospongin C and GPN were obtained from Biomol (Hamburg, DE). 7-Diethylamino-4-hydroxymethyl-coumarin was a kind gift from Rainer Müller (EMBL Heidelberg, DE). Deuterated solvents for NMR analysis were purchased from Deutero (Karlsruhe, DE). Sphingosine kinase inhibitor N-N-dimethyl-sphingosine was obtained from Sigma-Aldrich. NAADP-antagonist Ned-19 was purchased from Tocris Biosciences (Bristol, UK). The fluorescent calcium indicator Fluo-4-AM and Alexa488-azide were obtained from Life Technologies (Thermo Fisher Scientific, Waltham, USA). pEGFP-N1-TFEB was a gift from Shawn Ferguson (Addgene plasmid # 38119) (*Roczniak-Ferguson et al., 2012*).

### General synthetic procedures

All chemical reactions were carried out using dry solvents under inert atmosphere. Thin layer chromatography (TLC) was performed on plates of silica gel (Merck, 60 $F_{254}$) and visualized using UV light (254 nm or 366 nm) or a solution of phosphomolybdic acid in EtOH (10% w/v). HPLC grade solvents for chromatography were obtained from VMR. Preparative column chromatography was carried out using Merck silica gel 60 (grain size 0.063–0.200 nm) under a pressure of <1.5 bar. $^1$H-NMR spectroscopic measurements were conducted on a 400 MHz Bruker UltraShield™ spectrometer at 25°C. $^{13}$C-NMR measurements were performed on a 500 MHz Bruker UltraShield™ spectrometer at 25°C and were broadband hydrogen decoupled. Chemical shifts are given in ppm referenced to the residual solvent peak. *J* values are given in Hz and splitting patterns are designated using s for singlet, d for doublet, t for triplet, q for quartet, m for multiplet and b for broad signal. High-resolution mass spectra were recorded at the Organic Chemistry Institute of the University of Heidelberg

### Synthesis of 7-(diethylamino)-coumarin-4-yl)-methyl-chloroformate

A solution of 7-diethylamino-4-hydroxymethylcoumarin (*Schönleber et al., 2002*) (48 mg, 194 µmol) in 2 mL dry THF was cooled to 0°C. DIPEA (0.1 µL, 575 µmol) and phosgene (300 µL, 610 µmol) were added dropwise and stirred in the dark for 2 h at 0°C. The reaction mixture was extracted with EtOAc/$H_2O$ (1:1, 75 mL), the layers were separated, the organic layer was washed with brine and dried using $Na_2SO_4$. The solvent was removed under reduced pressure and the product was dried

further under high vacuum conditions. 7-(Diethylamino)-coumarin-4-yl]-methyl chloroformate was used without further purification.

## Synthesis of *N*-(7-(diethylamino)-coumarin -4-yl)-methyl)- ((2R,3S,E)-1,3-dihydroxyoctadec-4-en-2-yl)carbamate (Sph-Cou)

To a solution of D-erythro-sphingosine (30 mg, 100 µmol) in 2 mL dry THF, TEA (70 µL, 500 µmol) and a solution of 7-(diethylamino)-coumarin-4-yl)-methyl chloroformate (46 mg, 148 µmol) in 1 mL dry THF were added. The mixture was stirred at RT for 1 h in the dark. EtOAc (50 mL) was added to stop the reaction and the mixture was washed twice with citric acid (5% w/v, 25 mL) and twice with saturated NaHCO$_3$. The organic layer was dried with Na$_2$SO$_4$ and the solvent was removed under reduced pressure. The residue was purified by repeated flash chromatography (first column: eluent: DCM/MeOH13:1, second column: eluent: cyclohexane/EtOAc 1:5 (+1% TEA)). The compound **1** was obtained as yellow oil (24 mg, 42 µmol, 42% over two steps) $^1$H NMR (400 MHz, CDCl$_3$) $\delta$ = 7.29 (d, J=8.9, 1H), 6.58 (d, J=8.7, 1H), 6.50 (s, 1H), 6.14 (s, 1H), 5.96 (d, J=8.3, 1H), 5.80 (dd, J=14.6, 7.2, 1H), 5.55 (dd, J=15.2, 6.1, 1H), 5.22 (s, 2H), 4.39 (s, 1H), 4.05–3.96 (m, 1H), 3.76 (d, J=11.6, 1H), 3.68 (q, J=3.8, 1H), 3.40 (q, J=7.0, 3H), 3.06 (s, 1H), 2.05 (dd, J=13.5, 6.6, 2H), 1.41–1.10 (m, 28H), 0.87 (t, J=6.6, 3H). $^{13}$C NMR (126 MHz, CDCl$_3$) $\delta$ = 162.15, 156.18, 155.60, 150.40, 134.45, 128.70, 126.78, 124.43, 108.98, 106.16, 98.13, 74.75, 74.17, 66.85, 62.21, 61.89, 55.73, 44.96, 32.29, 31.93, 29.70, 29.67, 29.64, 29.51, 29.37, 29.22, 29.10, 22.70, 14.12, 12.40. HRMS for C$_{33}$H$_{53}$N$_2$O$_6^+$ calculated: 573.39036; found: 573.39027.

## Synthesis of *N*-(2,4-dimethoxy-6-nitrophenyl)-((2R,3S,E)-1,3-dihydroxyoctadec-4-en-2-yl)carbamate (Sph-NB)

A solution of D-erythro-sphingosine (30 mg, 100 µmol) in 2 mL dry THF and TEA (70 µL, 500 µmol) was stirred and 4,5-Dimethoxy-2-nitrobenzyl-chloroformate (41 mg, 150 µmol) in 2 mL dry THF was added dropwise. The mixture was stirred at RT for 1h in the dark. EtOAc (50 mL) was added and the mixture was washed twice with citric acid (5% w/v, 25 mL) and twice with saturated NaHCO$_3$. The organic layer was dried with Na$_2$SO$_4$ and the solvent was removed under reduced pressure. The residue was purified by flash chromatography (eluent: cyclohexane/EtOAc 1:5 (+1% TEA)). Compound **2** was obtained as colorless oil (51.4 mg, 95 µmol, 95% ) $^1$H NMR (400 MHz, CDCl$_3$) $\delta$ = 7.70 (s, 1H), 7.02 (s, 1H), 5.84–5.75 (m, 2H), 5.55 (d, J=6.0, 1H), 5.51 (s, 2H), 4.38 (s, 1H), 4.01 (s, 1H), 3.98 (s, 3H), 3.95 (s, 3H), 3.73 (d, J=11.5, 1H), 3.67 (s, 1H), 2.39 (s, 2H), 2.12–1.96 (m, 2H), 1.42–1.13 (m, 22H), 0.87 (t, J=6.7, 3H) $^{13}$C NMR (101 MHz, CDCl$_3$) $\delta$ = 155.99, 134.47, 128.58, 110.07, 108.15, 74.81, 63.78, 62.24, 56.52, 56.50, 56.40, 55.57, 32.28, 31.94, 31.92, 29.71, 29.68, 29.62, 29.48, 29.36, 29.22, 29.08, 22.71, 22.69, 14.13. HRMS for C$_{28}$H$_{47}$N$_2$O$_8$Na$^+$ calculated: 561.31519, found: 561.31557.

## Synthesis of *N*-(7-(diethylamino)-coumarin-4-yl)-methyl)-((2R,3S)-1,3-dihydroxyoctadecan-2-yl)carbamate (dhSph-Cou)

To a solution of D-erythro-dihydrosphingosine (10 mg, 33 µmol) in 1 mL dry THF, DIPEA (23 µL, 230 µmol) and a solution of [7-(diethylamino)-coumarin-4-yl)-methyl chloroformate (15 mg, 50 µmol) in 0,5 mL dry THF were added. The mixture was stirred at RT for 1.5 h in the dark. EtOAc (20 mL) was added to stop the reaction and the mixture was washed twice with citric acid (5% w/v, 10 mL) and twice with saturated NaHCO$_3$. The organic layer was dried with Na$_2$SO$_4$ and the solvent was removed under reduced pressure. The residue was purified by repeated flash chromatography (first column: eluent: DCM/MeOH 13:1, second column: eluent: cyclohexane/EtOAc 1:5). Compound **3** was obtained as yellow oil (14.5 mg, 25 µmol, 76% over two steps) $^1$H NMR (400 MHz, CDCl$_3$) $\delta$ = 7.27 (d, J=8.9 Hz, 1H), 6.56 (dd, J=9.0, 2.3, 1H), 6.49 (d, J=2.3, 1H), 6.14 (s, 1H), 6.08 (d, J=8.4, 1H), 5.22 (s, 2H), 4.07 (dd, J=11.4, 2.3, 1H), 3.82 (d, J=11.7, 2H), 3.66 (s, solvent THF), 3.62 (dd, J=8.1, 3.3, 1H), 3.40 (q, J=7.0, 4H),2,30 (t, J=7.6, solvent), 1.67–1.45 (m, 4H), 1.36–1.13 (m, 30H), 0.87 (t, J=6.7, 3H) $^{13}$C NMR (126 MHz, CDCl$_3$) $\delta$ = 162.34, 156.17, 155.52, 150.60, 130.02, 129.77, 124.41, 108.83, 106.12, 105.90, 97.90, 74.36, 62.24, 61.83, 55.13, 51.44, 44.82, 34.55, 31.94, 29.71, 29.67, 29.61, 29.59, 29.37, 27.23, 25.99, 24.96, 22.70, 14.12, 12.43. HRMS for C$_{33}$H$_{55}$N$_2$O$_6^+$ calculated: 575.40601; found: 575.40626.

## Cell culture and transfection

HeLa cells (LGC-ATCC, No. CCL-2, authenticated by STR profiling) were grown in DMEM (1 g/L Glutamate, Gibco) with 10% FBS (Gibco) and 1% Primocin (Invivogen), regularly tested for mycoplasma (using Lookout Mycoplasma PCR detection kit (Sigma, MP0035)) and only used when negative. Primary MEF from $Tpcn1^{-/-}$, $Tpcn2^{-/-}$ and double knock out mice were characterized previously (*Ruas et al., 2015*) and only used for 3-4 passages. They were not screened for mycoplasma. Human patient fibroblasts from MLIV patients were obtained from the Coriell Institute (MLIV patient cells: GM02525, authenticated by Nucleoside Phosphorylase Isoenzyme Electrophoresis, control cells: GM05399, authenticated by Chromosome Analysis) and checked for mycoplasma using the Lookout Mycoplasma PCR detection kit (Sigma, MP0035). In case of contamination, Mycoplasma Removal Agent (AbD Serotec, BUF035) was used. NPC1 cells were provided by Dr. Forbes D. Porter (NIHCD, USA). They are primary cultures from an NPC1 patient enrolled in the NIH program. The cells were not screened for mycoplasma, and used for 3–4 passages only. The culture medium for all mentioned fibroblasts consisted of DMEM (4.5 g/L glutatmate, Gibco) with 10% FBS (Gibco), 1% penicillin-streptomycin and 1% L-glutamine (Gibco). Cells were seeded in 8-well Lab-Tek™ microscope dishes 48–72 h prior to the experiment. If needed, cells were transfected in OPTIMEM (Gibco) with maximal 200 ng DNA per well and Fugene HD (Promega) according to the manufacturer's instructions. For the duration of the experiment, the medium of the cells was changed to imaging buffer (20 mM HEPES, 115 mM NaCl, 1.2 mM $MgCl_2$, 1.2 mM $K_2HPO_4$, 1.8 mM $CaCl_2$ and 10 mM glucose).

## Cell loading with caged compounds

Each Labtek-well was loaded with 100 µl of a 5 µM Fluo-4 AM (Molecular Probes®) solution in imaging buffer for 30 min at 37°C. The caged lipids were added to a final concentration of 2 µM 15 min prior to the experiment. Immediately prior to the experiment, the cells were washed and kept in imaging buffer.

## TLC analyses

Cells were grown to ~85% confluency in 6 cm dishes and pulsed with indicated concentrations of Sph-Cou in imaging buffer for the indicated times. Under +UV conditions, the cells were irradiated by a UV mercury arc source (Newport) equipped with a 400 nm highpass filter at a distance of 35 cm for 2 min. Cells were washed with PBS and scraped off in 300 µl PBS. The lipids were extracted according to a protocol published by Thiele et al. (*Thiele et al., 2012*). In brief, 600 µl methanol and 150 µl chloroform was added to the cells, vortexed and centrifuged at 14 000 rpm for 3 min. The supernatant was mixed with 300 µl chloroform and 600 µl acetic acid (0.1% in water). The aqueous phase was discarded and the organic phase was dried on a speedvac at 30°C for 15 min. The dried lipids were redissolved in 30 µl chloroform and applied on a 10 x 10 cm HPTLC Silica 60 glass plate (VWR) using the automatic Camaq system. TLC plates were developed using first $CHCl_3$/MeOH/$H_2O$/AcOH 65:25:4:1 for 6 cm and then cyclohexane/ethylacetate 1:1 for 9 cm. Fluorescently labeled lipids were visualized using a geldoc system.

## Lipidomic analyses

HeLa cells were grown in 6 cm dishes (Nunc) to 85–95% confluency and labelled with 2 µM Sph-Cou or 2 µM dhSph-Cou in imaging buffer or only in imaging buffer for 15 min. After washing with PBS, cells were transferred onto an ice block and UV-irradiated with > 400 nm light on a 450–1,000 W high-pressure mercury lamp (Newport, #66924, series #1166) for 2 min and then immediately pelleted (by centrifugation with 3000 rpm for 1 min) and snap-frozen in liquid nitrogen. Long chain base extraction and analysis was performed as follows. Cell pellets were resuspended in extraction solvent EtOH/$H_2O$/diethylether/pyridine (15:15:5:1) and ammonium hydroxide ($2.1.10^{-3}$ N). Cells were extracted by vortexing at 4°C for 10 min and incubation on ice for 20 min. Cell debris was pelleted by centrifugation at 20000 x *g* for 2 min at 4°C. The extraction was repeated one more time without the incubation on ice. Supernatants were combined and dried under vacuum in a Centrivap (Labconco, Kansas City, USA). The extract was resuspended in borate buffer (200 mM boric acid pH8.8, 10 mM tris(2-carboxyethyl)-phosphine, 10 mM ascorbic acid and 33.7 µM $^{15}N^{13}C$-valine) and derivatized by reaction for 15 min at 55°C with 6-aminoquinolyl-*N*-hydroxysuccinimidyl carbamate (AQC, 2.85 mg/ml in acetonitrile). The AQC reagent was synthesized according to Cohen et al (*Cohen and*

*Michaud, 1993*). Samples were analyzed after overnight incubation at 24°C using a reverse-phase C18 column (HPLC EC 100/2 Nucleoshell RP-18 2.7 µm) on an Accela system high-performance liquid chromatography (ThermoFisher Scientific, Waltham, MA), coupled to a TSQ Vantage (Thermo-Fisher Scientific, Waltham, USA). MRM-MS was used to identify and quantify lipid species. The relative amounts of long chain bases were normalized to a standard for the derivatization process ($^{15}N^{13}C$-valine) and internal standards added before extraction (C17-sphingosine, C17-sphinganine, C17-sphingosine-1-phosphate, C17-sphinganine-1-phosphate).

## Confocal time-lapse live cell microscopy and photoactivation

Fluorescence change of the calcium indicators were captured on a dual scanner confocal laser scanning microscope (Olympus Fluoview 1200) with a 63x oil objective using excitation at 488 nm and emission settings between 500–550 nm at an interval of 1 s per frame. For monitoring the red calcium dye R-GECO, the excitation was at 559 nm and emission between 580–650 nm. A baseline of 10 s was captured before photoactivation. Intracellular photoactivation was performed using the tornado function of the Olympus software with a circular region of interest (10 pixel units diameter, 8.9 µm²). For uncaging at 405 nm, we used 50% laser intensity for 3 s at 2 µs per pixel. Nitroveratroyl-uncaging at 375 nm was carried out using 100% laser intensity for 5 s at 2 µs per pixel.

## Data analysis

The acquired time lapse series were analyzed with the Fiji software (W. Rasband, NIH, USA) using the FluoQ macro (*Stein et al., 2013*) set to the following parameters:

Background subtraction method: Mean of an interactively selected ROI
Noise reduction / smoothing method: None
Threshold method: Interactively with ImageJ's built-in threshold window
ROI segmentation: Semi-automatically with binary mask modification
Calculate amplitude changes: Using maximum observed amplitude change

The maximum amplitude values were calculated by subjecting the raw traces to a central moving average. The maximal amplitude $x_{response}^{max}$ of these smoothed traces was used to calculate the amplitude change in percent $\%\triangle_x^{max}$ according to following formula:

$$\%\triangle_x^{max} = \frac{x_{response}^{max} - \overline{x}_{baseline}}{|\overline{x}_{baseline}|} * 100$$

The resulting intensity series/amplitude values represent mean values of whole cells and were further analyzed using R software (*Development Core Team, R, 2014*).

## Sph visualization and co-localization

Cells were seeded onto 11mm coverslips placed in wells of a 24-well plate and labeled with 4 µM pacSph in imaging buffer for 10 min. Cells were washed, overlaid with 1 mL imaging buffer and UV-irradiated on ice for 2.5 min using a 450–1,000 W high-pressure mercury lamp (Newport, #66924, series #1166) equipped with a glass filter to remove wavelengths below 345 nm (Newport, #20CGA-345), operated at 1,000 W. Cells were immediately fixed with MeOH at -20°C for 20 min. Not cross-linked lipids were extracted by washing 3x with 1 mL of CHCl₃/MeOH/AcOH 10:55:0.75 (v/v) at RT. Cells were then incubated with 50 µl of click mixture (1 mM ascorbic acid, 100 µM TBTA, 1 mM CuSO₄ and 2 µM Alexa 488 azide in PBS) for 1 h at RT in the dark. Cells were then washed with PBS and incubated with 50 µl of primary antibody (rabbit α-LAMP1, Cell Signaling, 1:100 in PBS supplemented with 4% BSA and 0.02% Triton) overnight at 4°C. Coverslips were again washed in PBS and incubated with secondary antibody (α-rabbit conjugated to AlexaFluor555, Cell Signaling, 1:800) for 1 h, washed and mounted in DAPI-containing mounting medium (Vectashield, Vector Laboratories, Inc. Burlingame, CA 94010, #H-1200). Microscopy images were captured at RT using a confocal laser scanning microscope (Zeiss LSM780) with a 63x oil objective. Settings were as follows: DAPI-channel (405 nm excitation (ex), 409–475 nm emmission (em); green channel: 488 nm ex, 489–550 nm em; red channel: 561 nm ex, 569–655 nm em). Images were further processed using imageJ (http://rsb.info.nih.gov/ij/).

## Acknowledgements

We are grateful for the support by the staff of the Advanced Light Microscopy Facility of the European Molecular Biology Laboratory for maintaining the microscopes used in this study. We thank Frank Stein for helpful suggestions for data analysis with FluoQ. We gratefully acknowledge help from Isabelle Riezman for the long chain base analysis and support from the Swiss National Science Foundation and the NCCR Chemical Biology. F.M.P. is a Royal Society Wolfson Research Merit Award holder. The research leading to these results has received funding from the European Union Seventh Framework Programme (FP7 2007–2013) under grant agreement no 289278 - "Sphingonet".

## Additional information

### Funding

| Funder | Grant reference number | Author |
|---|---|---|
| Seventh Framework Programme | 289278 - Sphingonet | Doris Höglinger Howard Riezman Frances M Platt Carsten Schultz |
| National Center for Competence in Research | Chemical Biology | Auxiliadora Aguilera-Romero Howard Riezman |
| European Commission | EMBL Interdisciplinary Postdoc Programme | Per Haberkant |

The funders had no role in study design, data collection and interpretation, or the decision to submit the work for publication.

### Author contributions

DH, Conception and design, Acquisition of data, Analysis and interpretation of data, Drafting or revising the article; PH, FDP, AG, Drafting or revising the article, Contributed unpublished essential data or reagents; AAR, Design, performance and analysis of lipidomics experiments, Acquisition of data, Analysis and interpretation of data, Drafting or revising the article; HR, Design and analysis of lipidomics experiments, Analysis and interpretation of data, Drafting or revising the article; FMP, Conception and design, Drafting or revising the article, Contributed unpublished essential data or reagents; CS, Conception and design, Analysis and interpretation of data, Drafting or revising the article

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
