## [Decision Letter]

Thank you for submitting your work entitled "Sphingosine releases calcium from lysosomes via the two-pore channel 1" for peer review at *eLife*. Your submission has been favorably evaluated by Vivek Malhotra (Senior Editor), a Reviewing Editor, and two reviewers.

The reviewers have discussed the reviews with one another and the Reviewing editor has drafted this decision to help you prepare a revised submission.

The reviewers find the work potentially very exciting, but three issues need to be addressed. If you can address the first point convincingly, the reviewers felt that the story would be very important.

1) The main potential novelty is in defining a physiologic role for sph in regulation of calcium release from endolysosomes with consequent signaling and cell regulatory functions. This is highly suggested by this study, but the study relies primarily on a 'pharmacologic' albeit novel and powerful approach using the delivery of caged sph. As such, the role of endogenous sph is not established. In this regard, the 3.5 fold increase in intracellular sph after uncaging may represent a supraphysiologic response (one reviewer was not aware of physiologic conditions where intracellular sph increases by that much). The study would be much stronger if there were ANY additional results to implicate endogenous sph in this mechanism and this pathway.

Moreover, the authors use various approaches but they don't connect them or control for them appropriately. For example, they show that exogenous sph does not work but they don't measure cellular sph after the exogenous sph which is important. If this is at the same level as what they see with caged-sph, then they have a problem.

Similarly, if they are unable to uncage dh-sph (they don't measure free dh-sph in response to uncaging the way they measure sph after uncaging sph), then they cannot use this as a basis to claim high specificity (which is one of the most important points in support of the specificity and therefore significance of the sph effects on calcium.) The finding that dh-sph is not functional is a strong indication of specificity of action; therefore to clinch this point, the authors need to measure dh-sph after its uncaging.

2) There is no evidence that Sph directly interacts with the TPC1 channel-no targets of Sph were identified. They could use click chemistry and show that uncaged Sph is bound to something specific to a calcium transporter or something specific to the lysosome-but this is likely beyond the scope of the present study. Because exogenous Sph itself does not induce Ca^2+^ release, the Discussion, Conclusions, and the title should be toned down.

3) The effects of TFEB may or may not be related to the specific action of lysosomal sph. They use pac-sph to 'establish' the sub cellular localization of sph, but this compound would resemble more exogenous sph in its pharmacology and therefore should not act on TFEB or calcium. Does the pac-Sph translocate TFEB? If not, then its cellular localization is irrelevant. If it does, then how come 'plain' sph does not act on calcium? Does exogenous sph induce TFEB?

[Editors’ note: a previous version of this study was rejected after peer review, but the authors submitted for reconsideration. The first decision letter after peer review is shown below.]

Thank you for choosing to send your work entitled "Sphingosine releases calcium from endolysosomes via the two-pore channel 1" for consideration at *eLife*. Your full submission has been evaluated by Vivek Malhotra (Senior Editor) and three peer reviewers, one of whom is a member of our Board of Reviewing Editors, and the decision was reached after discussions between the reviewers. Based on our discussions and the comments below, we regret to inform you that the reviewers noted a number of issues that would need to be addressed in full before the manuscript could be considered further for presentation in *eLife*.

Sphingosine (SPH) has been long demonstrated to act as a bioactive lipid to mediate various biological responses, such as cell proliferation inhibition and programmed cell death, but the mechanisms for it to mediate biological responses remain largely unclear. The present manuscript makes an interesting and potentially important observation that SPH releases calcium from endolysosomes via the two-pore channel 1 (TPC1). The authors developed photoactivatable SPH, caged sph, which was taken up by cells and was converted to free SPH upon photoactivation. This allowed the authors to study how free SPH transiently activates the release of calcium from different cellular compartments. The authors provide evidence that free SPH on its own, not through its metabolite sphingosine-1-phosphate, can induce the release of calcium to the cytosol from intracellular calcium stores. More importantly, the authors demonstrate that the calcium release is specific to SPH but not to its naturally occurring analogue dihydrosphingosine that differs from SPH by a single double bond. These findings provide novel insights into the mechanism by which SPH mediates biological responses.

The results are incomplete with respect to biological effects resulting from calcium mobilization by SPH. One would predict that this magnitude of acute calcium release would mimic many of the effects of stimuli that induce downstream effects through calcium mobilization. Providing evidence with respect to some biological effects would significantly improve the manuscript.

It is worrying that sphingosine proper does not cause a Ca^2+^ signal (Figure 1—figure supplement 2). This really needs clarification. Additionally, the authors state in the Results and Discussion that Ca^2+^ waves initiate at the point of uncaging and propagate throughout the cell but this is not quantified and this is not clear from the supplemental movie.

The pharmacology of the sphingosine-evoked Ca^2+^ signals is not so well defined. In Figure 3, it is unclear exactly how Ca^2+^ influx was blocked (Ni v EGTA v both). The effect of XeC is modest but activation of TPCs is thought to result in secondary Ca^2+^ release from the ER. In Figure 3, GPN appears to induce a Ca^2+^ signal that is comparable in amplitude to ATP (which the authors ascribe to ER release) which is surprising. Perhaps the GPN response causes ER Ca^2+^ release. Uncaging after addition of ATP/GPN is complicated by the elevated baseline. Would exogenous ATP fully deplete ER stores of calcium? What about using thapsigargin? There is little discussion of why Ned-19 is ineffective. Please show effects of Ned-19 on NAADP action or some positive control (Figure 4—figure supplement 2).

It is clear in Figure 4 that sphingosine responses are reduced upon TPC1 knockdown but it is not so clear whether this is a direct effect of sphingosine on TPC1.

The authors claim that sphingosine-mediated Ca^2+^ signals are reduced in NPC (Figure 5). The quantification (Figure 5) does not reflect this (the proportion of responding cells is only slightly reduced) and it is unclear if there is an effect on the amplitude. The apparent reduction in signal in Figure 5 in NPC is not dissimilar to the effect of Sph kinase block in Figure 2 (FtY720), XeC/PM block in Figure 3 and TPC2 silencing in Figure 4 yet a role for the intended targets in sphingosine action were discounted in Figure 2-5.

The subcellular localisation/pulse chase data in Figure 6 needs quantitation and the authors' claim that sphingosine localizes to the endolysosomal system (Figure 6—figure supplement 1) is not so convincing.

The authors imply that reduced sphingosine-mediated Ca^2+^ signal in NPC is due to activation of TPC1 by elevated sphingosine levels causing a reduction of Ca^2+^ levels within the acid Ca^2+^ store. A previous study (led by co-authors) however concluded that reduced Ca^2+^ content in the acid Ca^2+^ store was due to reduced Ca^2+^ uptake (Lloyd-Evans, 2008). In the Introduction, the authors refer to a study where Ca^2+^ release from the acidic Ca^2+^ store was reduced in NPC but in that study (Shen; 2012), lysosomal Ca^2+^ content was unchanged and rather lipid accumulation was proposed to block Ca^2+^ release through TRPML. This is a little confusing and potentially misleading. Cause-effect relationships are blurred.

Other Points:

1) The approach to block the conversion of SPH into S1P with DMS and FTY720 is somewhat problematic because DMS is a non-specific SPHK inhibitor and FTY720 is a SPH analogue, and both may mimic SPH in inducing calcium release on their own. Please note that FTY FTY720 is not an inhibitor of SK1. Moreover, the concentrations of DMS and FTY may not be sufficient to displace sph from sphingosine kinase. The authors should use genetic approaches to block the conversion of SPH to S1P in cells

2) Results in Figure 1—figure supplement 1 do not establish metabolic inertness of the caged compound. There is time dependent uptake, the signal is too faint to detect a sizable metabolite in the order of 10-20% or so, and this approach does not evaluate for spontaneous uncaging (the product would not be fluorescent).

3) The authors need to measure levels of sphingosine and metabolites at the earliest time point if they are able to.

4) Results and Discussion; full name of EGTA is not ethylene tetra-acetic acid

5) Figure 3. It looks like PM block has a significant effect on the sph response? This appears to be of the same magnitude as the loss in NPC. Also, this figure, in contrast to the legend, does not show the effects of EGTA or Ni.

6) SPH can also be quickly converted to ceramides through ceramide synthases or reverse action of ceramidases. The authors should investigate if blocking the conversion of SPH to ceramides can affect the calcium release by uncaged SPH. Also, studies in Farber's disease would be important.

7) The authors should investigate if increasing the generation of endogenous SPH by activating a ceramidase can induce the release of calcium from a calcium store(s).

8) The authors could measure the content of calcium in the endolysosomal system to determine if uncaging SPH indeed empties this calcium store. This will give another line of evidence that SPH releases calcium from this particular cellular compartment.

9) The max. amplitude change comparisons are often hard to compare – would additional kermel density estimations help, or report means where the data are relatively Gaussian?

10) The data on localization of Sph in NPC are hard to interpret (Figure 6). Only one cell is shown, and it appears to include significant ER staining. LAMP1 staining also seems unusual in this one cell compared to other published reports for NPC1 mutant cells. Please add additional examples and consider including an ER marker to strengthen your conclusions.

11) A recent paper in Science showed a connection between NPC1 and TPC proteins in that both seem to be needed for Ebola virus entry. Please cite and speculate how your work relates to that story.

---

## [Author Response]

*1) The main potential novelty is in defining a physiologic role for sph in regulation of calcium release from endolysosomes with consequent signaling and cell regulatory functions.*

We agree that one of the most exciting findings of our work is a potential role of Sph. Because the subject is calcium release, the reviewers (and also the readers) expect to see an effect similar to IP_3_-induced calcium release from the ER. We believe that the effect of Sph on intralysosomal calcium levels is more subtle yet not less important. As our results on monitoring TFEB translocation clearly demonstrate, simple elevation of intracellular calcium by uncaging calcium does not induce translocation. The major effect of Sph on calcium homeostasis therefore is not likely to be the cytosolic calcium increase, but rather the emptying of the lysosomal compartment. Consequently, chronically elevated levels of Sph in NPC cells are responsible for lowered lysosomal calcium levels due to their effect on TPC1 channels. The physiological role is therefore only observable in a pathologic situation. We do NOT demonstrate a classical second messenger system where physiological Sph levels are elevated rapidly and a rapid response results.

*This is highly suggested by this study, but the study relies primarily on a 'pharmacologic' albeit novel and powerful approach using the delivery of caged sph.*

Based on what we said above, we now indeed create a potentially artificial Sph transient. However, we need to do this, because the global elevation of Sph above a certain level (10 µM) is toxic to cells. In NPC but not in WT cells, Sph accumulates (1-2 µM) in the lysosome, but not in other organelles.

*As such, the role of endogenous sph is not established. In this regard, the 3.5 fold increase in intracellular sph after uncaging may represent a supraphysiologic response (one reviewer was not aware of physiologic conditions where intracellular sph increases by that much).*

Here we run into one of the aforementioned misunderstandings. We failed to make it clear that the 3.5-fold increase in Sph levels after uncaging as measured by mass spectrometry was due to a supramaximal illumination for chemical analysis and higher than that achieved in the cell biology experiments. For the mass spec analysis we had to perform a batch experiment where a 6 cm dish was fully illuminated for 2 min to be able to detect the uncaged species. In the live cell microscopy experiments, we illuminate only a region of 9 µm2 within a single cell for 3 sec, before we start monitoring cytosolic calcium levels in the entire cell. Hence, the increase in Sph levels will much smaller in the physiological experiments (unfortunately, there is no fluorescent sensor available to directly measure Sph levels in cells).

The study would be much stronger if there were ANY additional results to implicate endogenous sph in this mechanism and this pathway.

We would like to state that there is indeed support of our claim that Sph acts as a signaling molecule in lysosomes. In an admittedly correlative study, Fran Platt’s lab showed that increased *endogenous* levels of Sph correlate well with reduced lysosomal calcium levels in NPC disease (Lloyd-Evans et al. Nat. Med. 2008).

Further, we have invested quite some effort to measure lysosomal calcium release on isolated lysosomes upon addition of “endogenous” Sph. However, after consultation of specialists in the field we learned that this has never been published before. Not surprisingly we failed to set up a robust assay that demonstrates the release of lysosomal calcium after Sph elevation in a lysosomal preparation. Measuring lysosomal calcium directly in living cells is complicated by the fact that all calcium sensors are strongly pH-dependent. Ideally, we would generate a new construct of TPC1 fused to an extralysosomal calcium indicator such as a genetically-encoded calcium indicator. This is subject of current investigations in the Galione lab and needs significant optimization before it will be useful for our purpose.

*Moreover, the authors use various approaches but they don't connect them or control for them appropriately. For example, they show that exogenous sph does not work but they don't measure cellular sph after the exogenous sph which is important. If this is at the same level as what they see with caged-sph, then they have a problem.*

We refer to the misunderstanding mentioned above. The levels of Sph measured in mass spec studies after uncaging is not relevant for the live-cell microscopy experiments. It simply served to show that Sph was not released until illumination and that “endogenous” Sph was released. Nevertheless, we tried to measure intracellular Sph levels after addition of exogenous Sph as suggested. However, this is quite challenging as most of the exogenous Sph simply associates with the PM and cannot be washed away. This has been previously seen by the Riezman lab when radiolabeling cells using 3H-sphinganine. This rapid cell association leads to overestimation of the “internal” Sph levels. A crude estimate of how much Sph was taken up by the cells can be gained from measuring S1P-levels, which increased quite slowly in a time-dependent manner (much slower than the calcium signal happens after uncaging Sph, see inserts). We are aware that a quantification of Sph after addition to cells is not useful because excess Sph cannot be washed away. Due to these complications, we decided not to include the data in the manuscript, but we add it for the reviewers only.

Author response image 1.Comparative lipid analysis by mass spectrometry.Lipid extracts of HeLa cells pulsed with (**A**). Sph-Cou and then uncaged or (**B**). upon addition of exogenous Sph were AQC derivatized and measured on a TSQ Vantage mass spectrometer. Values are normalized to the C17 internal standards.**DOI:**
http://dx.doi.org/10.7554/eLife.10616.019

*Similarly, if they are unable to uncage dh-sph (they don't measure free dh-sph in response to uncaging the way they measure sph after uncaging sph), then they cannot use this as a basis to claim high specificity (which is one of the most important points in support of the specificity and therefore significance of the sph effects on calcium.) The finding that dh-sph is not functional is a strong indication of specificity of action; therefore to clinch this point, the authors need to measure dh-sph after its uncaging.*

Again, this is a point we were not sufficiently clear about: In Figure 2, all controls were performed by uncaging dh-Sph. When we uncaged Sph in Figure 1—figure supplement 2, we also measured dh-Sph (which did not change). We now added to the data set the results of uncaging dh-Sph and measuring dh-Sph and Sph levels (Figure 1—figure supplement 2) and added a paragraph in the Results section.

*2) There is no evidence that Sph directly interacts with the TPC1 channel-no targets of Sph were identified. They could use click chemistry and show that uncaged Sph is bound to something specific to a calcium transporter or something specific to the lysosome-but this is likely beyond the scope of the present study. Because exogenous Sph itself does not induce Ca^2+^ release, the Discussion, Conclusions, and the title should be toned down.*

We agree that showing direct interaction of Sph with TPC1 would improve the manuscript and simplify conclusions. However, we currently only have hints pointing to such an interaction. Crosslinking experiments followed by proteomic profiling with pacSph (see Haberkant et al, 2015) revealed TPC1 as a Sph interactor in one experiment out of three (interestingly, TPC2 did not show up at all). Since TPC1 expression levels are quite low and any potential interaction could easily be missed in such a complex experiment, we tried crosslinking on lysosomal purifications and detection by Western Blot. Unfortunately, we could not reliably detect any such interaction. We also tried to overexpress a new TPC1-FLAG construct followed by crosslinking with pacSph. Again, Western Blot experiments were not successful with this construct. We are now considering the possibility that Sph, analogous to NAADP, exerts its function on TPC1 via an adaptor protein. The search for such proteins will unfortunately exceed the scope of this manuscript (as mentioned by the reviewers). However, we believe that the lysosomal interactome of Sph will be a fabulous topic for the future. To reflect these difficulties, we changed the title to “Intracellular sphingosine releases calcium from lysosomes”

*3) The effects of TFEB may or may not be related to the specific action of lysosomal sph. They use pac-sph to 'establish' the sub cellular localization of sph, but this compound would resemble more exogenous sph in its pharmacology and therefore should not act on TFEB or calcium.*

We acknowledge that there was a mix-up of results and reasoning on our side. We have now changed the first sentence subsection “Subcellular localization of Sph in NPC disease to “Next, we investigated whether Sph is localized to the endosomal/lysosomal compartment in control as well as NPC1 fibroblasts by using a newly reported photoactivatable and clickable version of Sph (pacSph) “We clearly demonstrate that pac-Sph is located predominantly to the lysosome. This is supported now by Haberkant et al. where the metabolism of pac-Sph is shown, albeit in cells that lack the S1P-lyase.

Does the pac-Sph translocate TFEB? If not, then its cellular localization is irrelevant. If it does, then how come 'plain' sph does not act on calcium? Does exogenous sph induce TFEB?

We investigated the effects of pacSph as well as exogenous Sph on TFEB and did not observe any translocation. This is to be expected as both lipids do not release lysosomal calcium. As for using pacSph to probe for lipid localization, we would argue that this is an important tool. Our study (and the related manuscript Haberkant et al, 2015) uses pacSph for the first time to visualize subcellular sphingolipid distribution. Even though pacSph does not induce lysosomal calcium, most probably because it does not reach sufficiently high concentrations in a short period of time, this probe still reports on endogenous Sph localization. This is important for our conclusion that Sph acts on the lysosome.

To demonstrate the difference of elevated Sph levels after uncaging vs addition, we now used a caged version of pac-Sph and showed that it elevates calcium levels after uncaging just as caged-Sph does (see the Figure 10).

Author response image 2.Uncaging of caged pacSph increases cytosolic calcium.Mean Fluo-4 fluorescence traces of HeLa cells incubated with Sph-Cou or TFS. Uncaging was performed in a small circular area within the cell for 3 s as indicated by the black bar. The standard error of the mean is plotted as error bars.**DOI:**
http://dx.doi.org/10.7554/eLife.10616.020

We think however that this additional data set inflates the manuscript and provide it only for inspection by the reviewers.

[Editors’ note: a previous version of this study was rejected after peer review, but the authors submitted for reconsideration. The first decision letter after peer review is shown below.]

*The results are incomplete with respect to biological effects resulting from calcium mobilization by SPH. One would predict that this magnitude of acute calcium release would mimic many of the effects of stimuli that induce downstream effects through calcium mobilization. Providing evidence with respect to some biological effects would significantly improve the manuscript.*

Following the findings of Medina et al. (Nat Cell Biol 2015), starvation conditions lead to mTOR release from the lysosomal surface as well as lysosomal calcium release, subsequent TFEB dephosphorylation and nuclear translocation. We confirm this now by showing that Sph-induced lysosomal calcium release leads to TFEB translocation (new Figure 7). Interestingly, uncaging calcium in the cytosol by a flash of light does not lead to TFEB translocation despite the same massive signal (new Figure 7). It should be mentioned that many other calcium effects will likely also be induced by lysosomal calcium release, simply because the fast increase following Sph uncaging will induce calcium-induced calcium release from the ER.

*It is worrying that sphingosine proper does not cause a Ca^2+^ signal (Figure 1—figure supplement 2). This really needs clarification.*

Unfortunately, the addition of Sph to the extracellular space is cytotoxic and frequently induces apoptosis. Therefore only low micromolar amounts can be added in a live cell experiment which is mentioned in several papers (e.g. Chang, J. Dermat. 2004). We believe that, while the lipid enters properly, fairly rapid metabolism will prevent that sufficient amounts will reach the lysosome. Local uncaging at internal membranes seems to overcome this problem. Sph is now available in sufficient amounts close to its target. Further, we can dose Sph release well by choosing a larger or smaller area or less light. We now discuss this in the main text.

*Additionally, the authors state in the Results and Discussion that Ca^2+^ waves initiate at the point of uncaging and propagate throughout the cell but this is not quantified and this is not clear from the supplemental movie.*

To us, it was not surprising that a calcium wave starts from the point of uncaging. This can be explained by the above mentioned calcium-induced calcium release that is known to propagate through the cytosol. In addition, the diffusion or transport of the released Sph will also not be as fast as a water soluble compound would be. Nevertheless, we eliminated the sentence stating that the calcium wave propagates through the cell to avoid confusion.

*The pharmacology of the sphingosine-evoked Ca^2+^ signals is not so well defined. In Figure 3, it is unclear exactly how Ca^2+^influx was blocked (Ni v EGTA v both).*

We thank the reviewers to spot this shortcoming. We now altered the figure legend accordingly.

*The effect of XeC is modest but activation of TPCs is thought to result in secondary Ca^2+^ release from the ER.*

First, it was important to see that XeC was unable to block Ca^2+^ release. This supported our hypothesis that Sph does not activate release from the ER. The traces shown in Figure 3 further show that the response to Sph uncaging is now reduced which we attribute to the lack of calcium-induced calcium release via the IP3 channel. It should be mentioned that the initial phase of the calcium increase is the same in XeC and control treated cells (and also in PM blocked cells).

*In Figure 3, GPN appears to induce a Ca^2+^ signal that is comparable in amplitude to ATP (which the authors ascribe to ER release) which is surprising. Perhaps the GPN response causes ER Ca^2+^ release.*

First, it was important to see that XeC was unable to block Ca^2+^ release. This supported our hypothesis that Sph does not activate release from the ER. The traces shown in Figure 3 further show that the response to Sph uncaging is now reduced which we attribute to the lack of calcium-induced calcium release via the IP3 channel. It should be mentioned that the initial phase of the calcium increase is the same in XeC and control treated cells (and also in PM blocked cells).

*Uncaging after addition of ATP/GPN is complicated by the elevated baseline. Would exogenous ATP fully deplete ER stores of calcium?*

We tried to give another dose of ATP and most cells did not produce a second calcium transient. Of course, this only controls for ATP dosage. The proper experiment would have been to over-express the adrenergic receptor. Nevertheless, the strong calcium release after ATP will have likely emptied most of the available ER pool. In fact, after ATP addition, Fluo-4 (K_d_ for Ca^2+^ = 345 nM) saturates meaning that the intracellular calcium concentration probably exceeds 1 µM. As ATP is not washed away, the ER stores should not refill.

*What about using thapsigargin?*

We tried thapsigargin once and found no calcium elevation after Sph uncaging. When looking up the literature, we found that thapsigargin-induced calcium mobilization indeed abolishes the effect of GPN, meaning that lysosomal calcium stores are sensitive to thapsigargin and likely its inhibitory effect on ATPases (Sivaramakrishnan et al., J. Cell Sci. 2012).

*There is little discussion of why Ned-19 is ineffective. Please show effects of Ned-19 on NAADP action or some positive control (Figure 4—figure supplement 2).*

Our mistake: It was shown that Ned-19 is ineffective on TPC1 (but triggers TPC2). We now added a sentence stating this fact and cite Pitt et al., Sci Signal 2014.

*It is clear in Figure 4 that sphingosine responses are reduced upon TPC1 knockdown but it is not so clear whether this is a direct effect of sphingosine on TPC1.*

We agree with the reviewers that we do not show direct binding of Sph to TPC1 and that this would be very important. But with all respect, this is a major task beyond the scope of this manuscript. We are aiming on showing this interaction in a future collaboration with structural biologists. For the reviewers only: we used a photo-crosslinking and clickable Sph derivative and performed proteomic analysis after cross-linking, clicking on biotin and extracting the lipid-protein conjugates. In one of two replicates, TPC1 was identified while TPC2 was not found. While these data are very preliminary, they support a direct interaction of Sph with TPC1.

*The authors claim that sphingosine-mediated* Ca^2+^
*signals are reduced in NPC (Figure 5). The quantification (Figure 5) does not reflect this (the proportion of responding cells is only slightly reduced) and it is unclear if there is an effect on the amplitude.*

The average responder shows a clearly reduced amplitude. Further, the fraction of cells that respond with less than 50% amplitude change was 28% in control cells and 69% in NPC fibroblasts.

*The apparent reduction in signal in Figure 5 in NPC is not dissimilar to the effect of Sph kinase block in Figure 2 (FtY720), XeC/PM block in Figure 3 and TPC2 silencing in Figure 4 yet a role for the intended targets in sphingosine action were discounted in Figure 2-5.*

We agree that we show many traces with a reduced calcium response after treatment. Here, however, there is no treatment. We just compare two very similar cell types (healthy vs NPC subject cells). Just because a pharmacological treatment reduces the calcium response, this does not invalidate a comparison of the two cell lines.

*The subcellular localisation/pulse chase data in Figure 6 needs quantitation and the authors' claim that sphingosine localizes to the endolysosomal system (Figure 6—figure supplement 1) is not so convincing.*

We thank the reviewers for this hint. We now quantified the colocalization by determining Pearson’s correlation coefficients, added a new Figure 6 and expanded the corresponding Figure 6—figure supplement 1.

*The authors imply that reduced sphingosine-mediated Ca^2+^ signal in NPC is due to activation of TPC1 by elevated sphingosine levels causing a reduction of Ca^2+^ levels within the acid Ca^2+^ store. A previous study (led by co-authors) however concluded that reduced Ca^2+^ content in the acid Ca^2+^ store was due to reduced Ca^2+^ uptake (Lloyd-Evans, 2008). In the Introduction, the authors refer to a study where Ca^2+^ release from the acidic Ca^2+^ store was reduced in NPC but in that study (Shen; 2012), lysosomal Ca^2+^ content was unchanged and rather lipid accumulation was proposed to block Ca^2+^ release through TRPML. This is a little confusing and potentially misleading. Cause-effect relationships are blurred.*

We agree that the statements are a bit confusing. We now changed the Introduction and made clear in the Discussion that the given statements are part of an ongoing discussion in the field.

*Other Points: 1) The approach to block the conversion of SPH into S1P with DMS and FTY720 is somewhat problematic because DMS is a non-specific SPHK inhibitor and FTY720 is a SPH analogue, and both may mimic SPH in inducing calcium release on their own. Please note that FTY FTY720 is not an inhibitor of SK1. Moreover, the concentrations of DMS and FTY may not be sufficient to displace sph from sphingosine kinase. The authors should use genetic approaches to block the conversion of SPH to S1P in cells.*

Unfortunately, the use of RNAi against both Sph kinases was unsuccessful, probably because we blocked the main sphingolipid degradation pathway leading to toxic sphingolipid levels (now stated in the text). We then employed the non-sphingosine analog inhibitor SKI-II which gave a similarly small effect on Sph-induced calcium release. The data is incorporated into Figure 2, Figure 2—figure supplement 2.

*2) Results in Figure 1—figure supplement 1 do not establish metabolic inertness of the caged compound. There is time dependent uptake, the signal is too faint to detect a sizable metabolite in the order of 10-20% or so, and this approach does not evaluate for spontaneous uncaging (the product would not be fluorescent).*

*3) The authors need to measure levels of sphingosine and metabolites at the earliest time point if they are able to.*

We agree that a thorough control of sphingosine levels before and after uncaging was missing. We now teamed up with the group of Howard Riezman. They performed a lipidomics analysis. Results: incubation with caged Sph does not lead to changes in any of the sphingoid bases (Figure 1—figure supplement 2). After uncaging, Sph increased 3.4-fold while dihydroSph did not.

*4) Results and Discussion; full name of EGTA is not ethylene tetra-acetic acid.*

Corrected, thank you.

*5) Figure 3. It looks like PM block has a significant effect on the sph response? This appears to be of the same magnitude as the loss in NPC. Also, this figure, in contrast to the legend, does not show the effects of EGTA or Ni.*

It is well known that the use of calcium channel blockers reduces the size of the intracellular calcium pool simply because the leaky ER loses calcium that is partially pumped out and cannot be refilled. Therefore, in PM blocked cells, the response will be always reduced.

*6) SPH can also be quickly converted to ceramides through ceramide synthases or reverse action of ceramidases. The authors should investigate if blocking the conversion of SPH to ceramides can affect the calcium release by uncaged SPH. Also, studies in Farber's disease would be important.*

Although we did not specifically investigate ceramide formation, the Sph kinase inhibitor FTY720 we used also happens to block ceramide synthase activity (Lahiri et al, J. Biol. Chem. 2009). Regarding Farber’s disease: there are around 80 cases reported worldwide and it would be very tough to get cells from such patients. We agree that it would be very interesting to do such experiments.

*7) The authors should investigate if increasing the generation of endogenous SPH by activating a ceramidase can induce the release of calcium from a calcium store(s).*

We are not aware of a small molecule able to activate ceramidase. Over-expression looks like a dirty experiment as the cell would try to adapt to the higher Sph levels, especially considering the aforementioned toxicity.

*8) The authors could measure the content of calcium in the endolysosomal system to determine if uncaging SPH indeed empties this calcium store. This will give another line of evidence that SPH releases calcium from this particular cellular compartment.*

This would be a great experiment. Unfortunately, these are also very difficult experiments as the lysosome is quite acidic (and worse, inhomogeneous ‘pH’) and calcium indicators will have altered calcium affinities (and quantification) under these conditions. Genetically encoded calcium sensors cannot be employed because the uncaging procedure would bleach them and hence would prevent quantification.

*9) The max. amplitude change comparisons are often hard to compare – would additional kermel density estimations help, or report means where the data are relatively Gaussian?*

We agree that the histograms are not very easy to compare. However, once the reader has focused on this form of presentation, it is quite informative. We looked at a Gaussian presentation of the mean. Unfortunately, the cell number available does not suffice to present decent curves.

*10) The data on localization of Sph in NPC are hard to interpret (Figure 6). Only one cell is shown, and it appears to include significant ER staining. LAMP1 staining also seems unusual in this one cell compared to other published reports for NPC1 mutant cells. Please add additional examples and consider including an ER marker to strengthen your conclusions.*

We agree. We now show at least two cells in all co-localization conditions. Further, we also show a new Figure 6 that provides the Pearson’s coefficient for 6 to 10 cells per co-localization.

11) A recent paper in Science showed a connection between NPC1 and TPC proteins in that both seem to be needed for Ebola virus entry. Please cite and speculate how your work relates to that story.

The mentioned paper came out after we submitted the original manuscript. We now cite this paper and discuss its content in the conclusion section.